## Registered report

psychology/cognition/behaviour

error detection, error processing, performance monitoring, motor inhibition

**Authors for correspondence:**
Anna Foerster
e-mail: anna.foerster@uni-wuerzburg.de
Roland Pfister
e-mail: roland.pfister@uni-wuerzburg.de

# Error cancellation

Anna Foerster[1], Marco Steinhauser[2],
Katharina A. Schwarz[1], Wilfried Kunde[1] and
Roland Pfister[1]

[1]Julius-Maximilians-Universität Würzburg, Würzburg, Germany
[2]Catholic University of Eichstätt-Ingolstadt, Eichstätt, Germany

 AF, 0000-0002-8520-4672; MS, 0000-0001-6800-6036;
KAS, 0000-0001-8714-790X; WK, 0000-0001-6256-8011;
RP, 0000-0002-4429-1052

The human cognitive system houses efficient mechanisms to monitor ongoing actions. Upon detecting an erroneous course of action, these mechanisms are commonly assumed to adjust cognitive processing to mitigate the error's consequences and to prevent future action slips. Here, we demonstrate that error detection has far earlier consequences by feeding back directly onto ongoing motor activity, thus cancelling erroneous movements immediately. We tested this prediction of immediate auto-correction by analysing how the force of correct and erroneous keypress actions evolves over time while controlling for cognitive and biomechanical constraints relating to response time and the peak force of a movement. We conclude that the force profiles are indicative of active cancellation by showing indications of shorter response durations for errors already within the first 100 ms, i.e. between the onset and the peak of the response, a timescale that has previously been related solely to error detection. This effect increased in a late phase of responding, i.e. after response force peaked until its offset, further corroborating that it indeed reflects cancellation efforts instead of consequences of planning or initiating the error.

## 1. Introduction

Human errors can lead to drastic consequences for agents and their environment. It is therefore not surprising that human action control is governed by efficient mechanisms to monitor and regulate ongoing performance. These mechanisms ensure that the cognitive system detects erroneous actions readily and takes adaptive measures to steer future behaviour toward success [1–5]. Efficient error detection allows for swift error-correction responses, and it promotes adaptions for upcoming actions as documented by observations such as post-error slowing [6–9]. Converging results from event-related electroencephalography consistently yielded a negative deflection that peaks within only 100 ms after error commission, which has been related to error detection [10,11]. Computational models of performance

monitoring further support the proposed link between early detection and subsequent countermeasures to correct unforeseen consequences and to avoid errors in upcoming actions [12,13].

Here we argue that efficient error detection might not only target future behaviour, but it might even feedback immediately onto the erroneous action by cancelling ongoing motor activity as quickly as possible [14–18]. Unexpected events such as errors have indeed been proposed to trigger an orienting response followed by general motor inhibition [19], a process that has been proposed to underlie observations such as post-error slowing. However, motor inhibitory signals could also instil immediate cancellation of an ongoing erroneous action. The benefits of such countermeasures apply especially to real-world actions, which typically unfold over an extended timescale and multiple consecutive steps. For example, if an agent is about to throw a letter into the wrong mail box, or ring a bell on a door instead of turning on the light switch in a hall, an adaptive system would aim at cancelling the current course of action as soon as possible. Similar tendencies might even operate for more ballistic keypress responses that have commonly been employed to study error processing. Several observations point to this possibility, by indicating that errors come with reduced peak forces (PFs) and shorter response durations (RDs) than correct responses [15,20–22]. Whether these observations are indicative of active cancellation or whether they derive from differences during motor planning and initiation is an open question that the present study aimed to resolve.

Cancellation on such short notice as in the case of a keypress response poses a profound challenge to current models of performance monitoring by suggesting auto-corrective effects for erroneous actions on a timescale that has previously been assumed to capture initial error detection instead. We hypothesize that active error cancellation would lead to shorter RDs (i.e. the time from onset of a keypress to its release) of erroneous than correct responses even when matching erroneous and correct responses for several cognitive and behavioural surface parameters. For one, erroneous responses often have shorter response times (RTs) than correct responses [15,20,23]. To arrive at a plausible estimate of the hypothesized effect of error commission on RD, and to examine the role of the predicted RT differences for our main measure of RDs, we re-analysed data from previous work in which we had participants perform a speeded choice reaction task and recorded onsets as well as offsets of their keypress responses (Experiments 1 of [24,25]). We analysed RDs and RTs as a joint function of accuracy (correct versus error) and quartiles of the RT distribution. Figure 1 summarizes the key findings of these analyses (see *Pilot data* for detailed results).

The pilot analyses document a marked difference in the duration of correct and erroneous responses across the RT distribution, consistent with the hypothesized process of active error cancellation (figure 1a,c). Moreover, RDs of erroneous responses became smaller at longer RTs whereas RDs of correct responses were relatively stable across the RT distribution. Visual inspection of figure 1b further indicated consistent pre-error speeding [27,28], with particularly fast error responses (figure 1b,d). These results suggest that erroneous responses come with systematically reduced duration that cannot be explained in terms of differing RTs.

Inferring error cancellation from reduced RDs, however, poses the additional challenge that errors also tend to come with smaller PFs than correct responses [15,18,20–22]. While this observation of attenuated error responses might itself be consistent with the notion of active cancellation, it might also originate from differences in how errors and correct responses are generated. The observation of lower PFs for errors therefore cannot make a clear case for active error cancellation. Furthermore, if errors were already triggered with reduced force, low PFs would imply different biomechanical constraints for errors as compared with (on average) more forceful correct responses. We therefore aimed at overcoming this potential confound by matching responses for RTs and especially for PFs. Such matching parallelizes potential biases so that any remaining differences between errors and correct responses are difficult to explain without assuming error cancellation. Observing reduced RDs for errors even in the matched data would therefore provide convincing indication of active cancellation of erroneous responses.

The main study tackled this challenge by means of continuous force measurements as compared with the discrete distinction of response onsets and offsets used in the pilot data. We hypothesized that active error cancellation leads to shorter RDs of erroneous than correct responses even when matching for RT and when matching for PF (Hypothesis 1; table 1). We pre-registered that we would only infer active error cancellation if commission errors come with shorter RDs than correct responses for the unmatched datasets and for both matched datasets. Mixed results for this main hypothesis were planned to be followed up by Bayesian analyses and potentially additional sampling to either refute or support the current standard model without immediate error cancellation.

Moreover, we aimed at discerning whether error cancellation emerges in an early or a late phase of responding. RD comprises two clearly distinguishable time epochs of the force profile, namely from force onset to PF, and from PF to force offset. Early, anticipatory error cancellation would hold that effects of cancellation in RDs occur already when a response reaches its PF, whereas late, reactive cancellation

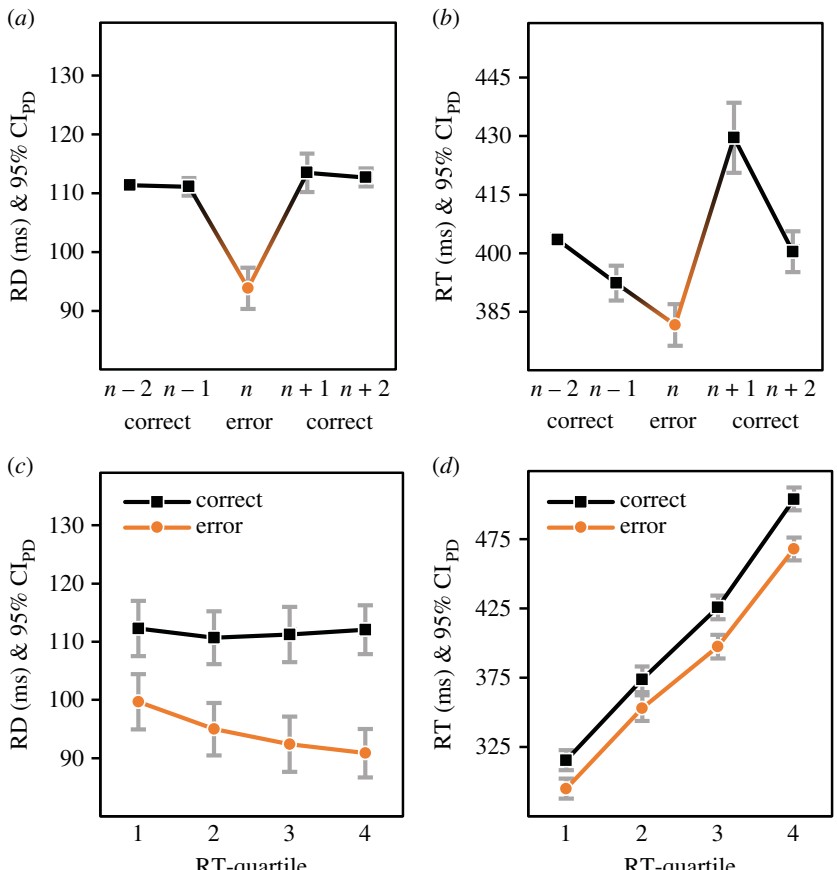

**Figure 1.** Mean RDs and RTs of the pilot analyses. (*a*) RDs and (*b*) RTs of trial sequences where an error (bright, orange dot) in trial *n* was preceded and followed by two correct responses (black squares), respectively. Error bars represent the 95% confidence interval for paired differences between consecutive trials ($CI_{PD}$; [26]). (*c*) RDs for each quartile of the RT distribution, with an increasing difference in RDs with RT level. Error bars indicate the 95% $CI_{PD}$ for each pairwise comparison of correct and erroneous responses. (*d*) Mean RTs for all RT distribution quartiles for correct and erroneous trials. Error bars indicate the 95% $CI_{PD}$ for each comparison of correct and erroneous responses.

would manifest only after the peak (Hypothesis 2). Continuous force measurements further allow for assessing whether erroneous responses are enacted with less overall force than correct responses in the unmatched and matched data, measured through the area under the curve (AUC) of the force profile (Hypothesis 3). In line with Hypothesis 2, we planned to test whether such attenuation is already present in an early phase of responding before reaching PF, or only emerges reactively afterward (Hypothesis 4). If the evidence pointed to an early reduction of force in the unmatched and both matched datasets, we planned to dissect the early force profile even further in relation to the registration of the response. For RT-matched data, we planned to scrutinize how the reduction of force develops for erroneous responses in five time steps leading up to the registration of the response (Hypothesis 5). In additional analyses, we aimed to assess whether overall stronger force reduction during erroneous relative to correct responding relates to the successful abortion and correction of subthreshold responses. In particular, we predicted that larger differences between correct and erroneous responding in AUCs would come with a higher percentage of low-threshold erroneous responses in correct trials of the unmatched data (Hypothesis 6). Finally, we planned to test whether we replicate shorter RTs (Hypothesis 7) and smaller PFs (Hypothesis 8) for erroneous than correct responses to validate our data against established findings in the literature.

## 2. Methods

The data and analysis code for the pilot analyses and for the pre-registered main study are publicly available, as is the data and analysis code for the validation study described in the electronic supplementary material and the Stage 1 protocol (osf.io/5v9es [30]). Table 1 provides an overview of our research questions, hypotheses, sampling plan, analysis plan and interpretation.

**Table 1.** Study design.

| question | hypothesis | sampling plan (e.g. power analysis) | analysis plan | interpretation given to different outcomes |
|---|---|---|---|---|
| Main question: does immediate error cancellation arise on the timescale of a (ballistic) keypress response? | (1) RDs are shorter for erroneous than correct responses. | We aimed for a high power of 99% to detect even the lower bound of $d_z = 1.34$, 95% $CI_d = [0.93; 1.74]$ of the differences between erroneous and correct responses in RDs from the pilot analyses. This power analysis suggested a sample size of at least 23 participants for a two-tailed paired $t$-test ($\alpha = 5\%$) and for the main effect of the factor accuracy in all analyses of variance (ANOVAs; see *Data analysis*). We recruited a total sample of 34 participants to allow for sufficient power also for all additional analyses (see following table rows). | Hypotheses (1) and (2) were tested in $2 \times 2$ ANOVAs with the within-subject factors accuracy (correct versus error) and time-window (pre-peak versus post-peak), and RDs as the dependent variable. We performed this analysis on unmatched data and on data where erroneous and correct responses were matched by RT and PF, respectively. The two analyses on the matched datasets were conditional upon the success of our (iterative) matching procedures, which we evaluated through equivalence tests ($-2$ ms $< \Delta$RTs $< 2$ ms; $-0.05$ a.u. $< \Delta$PFs $< 0.05$ a.u.). If equivalence tests did not indicate a sufficiently close match, we would re-run the matching algorithm on a trimmed dataset until reaching a satisfactory match on all equivalence tests. | We would only infer active error cancellation if commission errors came with shorter RDs than correct responses in all three analyses. If this comparison was non-significant in any of the three analyses, we would resort to a Bayesian approach and continue sampling until reaching a Bayes factor of $BF_{01} > 10$ or $BF_{01} < 0.1$ for all three analyses (using a Cauchy distribution with a scale parameter of 1 as prior). We would use separate Bayesian $t$-tests. If this outcome supported the null hypothesis of equal RDs across conditions, results would support the current standard model of detection without immediate cancellation. Observing significant effects on all three tests would overturn this model by showing immediate cancellation. In the case of a significant interaction between accuracy and time-window (i.e. evidence for differences in force attenuation between early and late phases of the response), we further explored the time course as per the following question. |

**Table 1.** (*Continued.*)

| question | hypothesis | sampling plan (e.g. power analysis) | analysis plan | interpretation given to different outcomes |
|---|---|---|---|---|
| Does the cancellation of ongoing erroneous actions emerge in early or late phases of responding? | (2) RDs are already shorter for erroneous than correct responses before the force profile of a response reaches its peak (PF). | | In the case of significant two-way interactions in the ANOVAs (see above), Hypothesis (2) was further tested in two one-tailed paired-samples t-tests. We tested for shorter RDs in erroneous than correct trials before and after the peak, respectively. | In the presence of a two-way interaction, we would only infer early error cancellation if there was a significant difference between the RDs of correct and erroneous responses for the pre-peak condition in all three analyses. Observing reduced RDs for errors only in the post-peak interval would support active but late error cancellation. |
| Are erroneous responses enacted with less overall force than correct actions? | (3) AUCs as computed for the force profile are smaller for erroneous than correct responses. | Effect sizes for comparisons of PFs (rather than overall force) in previous reports [15] point towards an effect of $d_z = 0.65$ for which about 33 participants allow for a power of 95%. Note that the effect size estimate can only be approximate, however, because the present study was the first to implement fine-grained analyses for force profiles rather than PF-based measures. The reported validation data suggested the estimate to be feasible (here: $d_z = 0.98$). | Hypotheses (3) and (4) were tested in $2 \times 2$ ANOVAs with the within-subject factors accuracy (correct versus error) and time-window (pre-peak versus post-peak) and AUCs as the dependent variable. We performed this analysis on unmatched data and on data where erroneous and correct responses were matched by RT and PF, respectively. The analysis on the matched data were again conditional upon the success of our (iterative) matching procedures (see above). | We would infer enactment of less overall force for erroneous than correct actions if commission errors came with lower AUCs than correct responses in all three analyses. In the case of a significant interaction between accuracy and time-window (i.e. evidence for differences in force attenuation between early and late phases of the response), we further explored the time course as per the following question. |

(*Continued.*)

**Table 1.** (*Continued.*)

| question | hypothesis | sampling plan (e.g. power analysis) | analysis plan | interpretation given to different outcomes |
|---|---|---|---|---|
| Does a weaker enactment of force in erroneous actions emerge in early or late phases of responding? | (4) AUCs are smaller for erroneous than correct responses already before a response reaches its PF. | | In the case of significant two-way interactions in the ANOVAs (see above), Hypothesis (4) was further tested in two one-tailed paired-samples *t*-tests. We tested for shorter AUCs in erroneous than correct trials before and after the peak, respectively. | In the presence of a two-way interaction, we would only infer a weaker enactment of force for erroneous than correct responses in an early phase if there was a significant difference between the AUCs of correct and erroneous responses for the pre-peak condition in all three analyses. Observing smaller AUCs only in the post-peak interval would support active but late attenuation of response force. |

**Table 1.** (Continued.)

| question | hypothesis | sampling plan (e.g. power analysis) | analysis plan | interpretation given to different outcomes |
|---|---|---|---|---|
| | (5) Mean forces in RT-matched data are smaller for erroneous than correct responses in the time course before response registration (i.e. mean matched RT). | | If all three pre-peak tests for Hypothesis (4) returned significant results, we would test Hypothesis (5) in a $5 \times 2$ ANOVA with the within-subjects factors time-window (99 to 80 ms versus 79 to 60 ms versus 59 to 40 ms versus 39 to 20 ms versus 19 ms to mean matched RT) and accuracy (correct versus error) and mean force as the dependent variable. In the case of a significant two-way interaction, Hypothesis (5) was further tested in a one-tailed paired-samples $t$-test per time-window to probe for smaller mean forces in erroneous than correct trials. | We would infer a weaker enactment of force for erroneous responses in all time-windows preceding the response if there was a significant main effect without a significant interaction in the omnibus ANOVA. In the case of a significant interaction, we confined this interpretation to time-windows with a significant difference between correct and erroneous responses. |
| Does a more pronounced attenuation of overall response force in erroneous relative to correct responses relate to the successful abortion of erroneous subthreshold responses in correct trials? | (6) Differences between erroneous and correct responses in AUCs ($\Delta_{AUC}$) correlate positively with the percentage of low-threshold erroneous responses in correct trials. | The positive correlation in the validation study was large with $r = 0.81$. A sample size of about 11 participants would already have a power of 95% to detect a correlation of $r = 0.81$ in a one-tailed correlational analysis ($\alpha = 5\%$; calculated with the *pwr.r.test* function of the R package *pwr* v. 1.3-0; [29]). We used the entire sample ($n = 34$) based on the lowest estimated effect size (see Hypothesis 3). | Hypothesis (6) was tested in a one-tailed Pearson-correlation between $\Delta_{AUC}$ and the percentage of low-threshold erroneous responses in correct trials. | We would infer that a stronger difference in the enactment of overall force in erroneous relative to correct responses relates to the successful abortion of erroneous subthreshold responses in correct trials if the correlation was significantly positive. In the case of a non-significant correlation, we computed a Bayes factor using a shifted, scaled beta distribution as prior ($r$ scale parameter = 1/3). |

(Continued.)

**Table 1.** (*Continued*.)

| question | hypothesis | sampling plan (e.g. power analysis) | analysis plan | interpretation given to different outcomes |
|---|---|---|---|---|
| Are erroneous actions initiated faster than correct responses? | (7) Erroneous responses have shorter RTs than correct responses. | Effect sizes of the differences between erroneous and correct responses in RTs of the pilot data and the validation study ($d_z \geq 1.06$) consistently exceeded the lower bound of the effect size considered for the power analyses of Hypotheses (1) and (2) ($d_z = 0.93$) so that the power for this analysis is well above 99%. | Hypothesis (7) was tested in a one-tailed paired-samples $t$-test (i.e. error < correct) with RTs as the dependent variable. | We inferred faster initiation of erroneous than correct responses in the case of a significant result. Careful matching of erroneous and correct trials for one of the three main analyses (as described in the first row of this table) would be performed even if there was a non-significant, numerical difference in RTs to be maximally conservative. |
| Are erroneous responses enacted with less maximum force than correct actions? | (8) Erroneous responses have lower PFs than correct responses. | See Hypotheses (3) and (4). | Hypothesis (8) was tested in a one-tailed paired-samples $t$-test (i.e. error < correct) with PFs as the dependent variable). | We would infer enactment of less maximum force for erroneous than correct actions if commission errors come with lower PFs. Careful matching of erroneous and correct trials for one of the three main analyses (as described in the first row of this table) would be performed even if there was a non-significant, numerical difference in PFs to be maximally conservative. |

## 2.1. Pilot data

We reanalysed a dataset with 48 participants of which we had to exclude four datasets because of insufficient observations in at least one design cell (see below). The design of the pilot study resembled the main study (see *Stimuli and apparatus* and *Procedure*) with the major exception that participants responded on a standard German QWERTZ keyboard instead of the custom-built apparatus to measure force profiles as used in the main study. In short, participants responded to target letters in a speeded choice reaction task with a $4:2$ mapping of target stimuli to response keys (see electronic supplementary material, figure S1A for the trial procedure). Four task-irrelevant distractor letters surrounded each target letter to increase perceptual noise and thus error likelihood. Participants worked on 1120 trials of this task in 20 blocks with the first block serving as practice.

For each response, we measured RT (i.e. time from target onset to keypress) and RD (i.e. time from pressing to releasing the key) to gather first evidence for active error cancellation (figure 1). We excluded the practice block and the first trial of each block. We selected trials with a correct response in the preceding trial (19.5% excluded). We then discarded trials with miscellaneous errors where participants used any other than the instructed keys or responded multiple times (2.0%), as well as omission errors (2.6%). We further excluded trials as outliers if either RT or RD deviated more than 2.5 s.d. from their cell mean, calculated separately for each participant and accuracy condition (2.3%). For visualization in figure 1a,b, we selected trial sequences with two correct responses preceding and following an error, respectively.

We compared correct and erroneous RTs and RDs in separate two-tailed paired-samples $t$-tests. In addition, we computed RT-quartiles (*ntiles* function of the R package *schoRsch* v. 1.9.1; [31,32]) and analysed RTs and RDs as a function of accuracy and RT-quartile in separate $2 \times 4$ analyses of variance (ANOVAs). In the case of significant two-way interactions, we tested for effects of accuracy in each quartile via two-tailed paired-samples $t$-tests. We excluded four participants from the analyses because they provided less than 10 observations in at least one of the design cells of the ANOVAs.

Erroneous responses indeed came with markedly shorter RDs than correct ones (94 versus 112 ms; figure 1a), $\Delta_{RD} = 17$ ms, $t_{43} = 8.91$, $p < 0.001$, $d_z = 1.34$, 95% CI = [13 ms, 21 ms], and the same held true for RTs (379 versus 405 ms; figure 1b), $\Delta_{RT} = 26$ ms, $t_{43} = 7.43$, $p < 0.001$, $d_z = 1.12$, 95% CI = [19 ms, 33 ms].[1] The distributional analysis replicated the observation of shorter RDs for erroneous as compared with correct responses (figure 1c), $F_{1,43} = 78.44$, $p < 0.001$, $\eta_p^2 = 0.65$, $d_z = 1.34$. This analysis further yielded shorter RDs with increasing RT-quartile, $F_{3,129} = 4.72$, $p = 0.014$, $\eta_p^2 = 0.10$ ($\varepsilon = 0.61$), and a significant interaction of both factors, $F_{3,129} = 7.66$, $p = 0.001$, $\eta_p^2 = 0.15$ ($\varepsilon = 0.69$). The effect of accuracy on RDs increased across RT-quartiles and it was significant for each individual quartile, $ts_{43} \geq 5.36$, $ps < 0.001$, $d_z \geq 0.81$ (95% $CI_{25} = [8$ ms, 17 ms], 95% $CI_{50} = [11$ ms, 20 ms], 95% $CI_{75} = [14$ ms, 24 ms], 95% $CI_{100} = [17$ ms, 25 ms]). In the distributional analysis of RTs, erroneous responses were faster than correct responses (figure 1d), $F_{1,43} = 56.45$, $p < 0.001$, $\eta_p^2 = 0.57$, $d_z = 1.13$, 95% CI = [19 ms, 33 ms], and RTs trivially increased across quartiles, $F_{3,129} = 678.44$, $p < 0.001$, $\eta_p^2 = 0.94$ ($\varepsilon = 0.38$). The interaction of both factors was significant, $F_{3,129} = 8.76$, $p < 0.001$, $\eta_p^2 = 0.17$ ($\varepsilon = 0.76$). Effects of accuracy increased for higher quartiles but were significant for all RT-quartiles, $ts_{43} \geq 4.54$, $ps < 0.001$, $d_z \geq 0.68$ (95% $CI_{25} = [13$ ms, 28 ms], 95% $CI_{50} = [12$ ms, 30 ms], 95% $CI_{75} = [20$ ms, 37 ms], 95% $CI_{100} = [28$ ms, 44 ms]).

## 2.2. Sample

For our main research question (Hypothesis 1), we computed the 95% $CI_d$ for the effect size $d_z = 1.34$, 95% $CI_d = [0.93; 1.74]$, as observed for the differences between RDs of erroneous and correct responses in the pilot analyses (*ci.sm* function of the R package *MBESS* v. 4.8.0; [33]). Because the planned matching procedures probably constrain the resulting effect size, we opted to use the lower bound of the

---

[1]In the pilot study, participants did not receive immediate feedback for correct responses and commission errors, but they received a summary of their performance after each block. We handled performance feedback similarly in the main study (see *Stimuli and apparatus* and *Procedure*). Validating this design choice, we found similar effects of error cancellation in a replication study of the reported work, where we manipulated between participants whether the commission of an error was or was not fed back at the end of a trial (see electronic supplementary material, figure S1 for the trial procedure; [25]). Again, we found shorter RDs, $F_{1,92} = 62.20$, $p < 0.001$, $\eta_p^2 = 0.40$, $d_z = 0.81$, and shorter RTs, $F_{1,92} = 106.37$, $p < 0.001$, $\eta_p^2 = 0.54$, $d_z = 1.06$, for errors than for correct responses across feedback conditions (with non-significant main effects of feedback and two-way interactions of accuracy and feedback in RDs and RTs, $Fs_{1,92} \leq 1.52$, $ps \geq 0.220$, $\eta_p^2 s \leq 0.02$, $d_z \leq 0.13$). We relied on the effect size estimate of the first pilot analyses for the power analysis of the main study, however, because feedback and temporal characteristics match the main study more closely than the replication design.

confidence interval as a conservative estimate. About 23 participants would ensure a high power of 99% to detect the effect size corresponding to this lower bound in a two-tailed test ($\alpha = 5\%$; *power.t.test* function of the R package *stats* v. 4.0.3; [32]). Considering the remaining hypotheses, the effect size for the difference between correct and erroneous responses in PFs informed from previous reports [20] was the smallest relevant effect size for any of the hypotheses tested in this study ($d_z = 0.65$). We thus considered this effect size for the computation of our sample size, and a small-scale validation of the proposed design ($N = 4$) found effects to exceed this estimate consistently across dependent measures and matching procedures (see the electronic supplementary material). We therefore tested 34 participants to arrive at a counterbalanced sample that ensured a high power of 95% in a two-tailed paired test for this hypothesis ($\alpha = 5\%$) while ensuring even higher power for all remaining hypotheses (including a power of more than 99% for our main hypothesis).

We planned to exclude the data of participants who opted to abort the study prematurely (which did not occur), for whom the study could not proceed as planned (see *Stimuli and apparatus* and *Procedure*) because of technical errors (four participants), and based on *a priori* criteria to establish sufficient data quality (see *Data selection*; six participants). We replaced excluded datasets with new participants until we had 34 datasets for statistical analyses. If the analyses for our main research question (Hypothesis 1) had returned mixed results, we would have increased the sample size by means of adaptive sampling informed by Bayes factors (see *Data analysis*).

## 2.3. Stimuli and apparatus

Stimuli were presented on a 24″ screen with a display resolution of $1920 \times 1080$ pixels and a refresh rate of 100 Hz. Participants responded with their two index fingers on custom-built keys that measured isometric force with a sampling rate of 250 Hz (see electronic supplementary material, figure S2). The force-sensitive parts of the keys had a size of $1.8 \times 1.8$ cm with an elevated circular platform (1.3 cm diameter) as finger rest. They were embedded in a frame that was $2.5 \times 2.5$ cm in width and depth and 1.4 cm in height. The combined apparatus of frame and the force-sensitive part was about 1.5 cm in height. Participants responded to four target letters, of which *T* and *N* mapped to one key while *V* and *K* mapped to the other key. The mapping of letter pairs to response keys was counterbalanced across participants. Target letters appeared in white font colour in the centre of a black screen closely surrounded by four distractor letters above, below (distance amounts to 6% of the screen height) and to both sides (3% of screen width).[2] The distractor letters were *O, W, X, U, Z, Y, H* and *A*. None of the distractors mapped onto a response but rather the presence of distractors increased visual noise to stimulate commission errors.

## 2.4. Procedure

Participants were explicitly instructed about the mapping rule before the study and they were encouraged to ignore the distractors that accompanied each target. The trial started with a fixation cross for 500 ms, followed by a display showing target and distractors for a maximum duration of 600 ms (target and distractors disappeared upon response onset). The force exerted on both response keys was measured from the onset of fixation. The first 10 measurements were averaged for each key and used as baseline. Response onsets were identified when the force on one key was at least 0.25 arbitrary units (a.u.; about 250 g or 2.5 newton) above its baseline. Previous experience with this device as well as a small validation study reported in the electronic supplementary material yielded a reasonable amount of omission errors with this response criterion while ensuring that the keys operate with sufficient sensitivity. RT then denoted the time from onset of the target until the force on one response key reached the response threshold. After the response onset had been registered, the screen went black for an inter-trial interval of 1000 ms while the force on both keys was still measured. The offset of the response was registered when the force matched or exceeded the threshold for the last time.

---

[2]In the accepted Stage 1 version of this article, we announced that distractors would appear in a distance of 3% screen height and 5% screen width from the target although we intended to use the same stimulus setup as in the validation study and in the pilot study (6% and 3%, respectively). We noticed our error during the preparation of data collection. We decided to implement our original stimulus arrangement, deviating from the preregistered method in this regard; however, we made this decision before any data had been collected.

Participants worked through one initial practice block, followed by 17 experimental blocks. In the practice block, participants received feedback for each response for 1000 ms at the end of the trial. Correct responses were fed back with 'Good!' (German: 'Gut!'), early responses during the fixation with 'Too early!' (German: 'Zu früh!'), commission errors (left response when right response would be appropriate and vice versa) with 'Wrong!' (German: 'Falsch!') and omissions of any response, i.e. no response before the deadline of 600 ms, with 'Too slow!' (German: 'Zu langsam!'). In experimental blocks, participants only received feedback for early responses and omissions. At the end of each block, participants received a summary on their performance with the mean RT of correct responses and the number of commission and omission errors. They were also urged to respond as quickly as possible while trying to avoid high numbers of errors independently from their performance in the block.

The practice block featured a random sequence of 32 trials in which each target appeared once with each distractor. In each experimental block, each combination of targets and distractors appeared twice, resulting in a random order of 64 trials.

## 2.5. Data selection

We neither analysed the practice block nor the first trial of each block. We then determined the frequency of error types for each participant and excluded participants who responded correctly in less than 60% of the trials (five participants; remaining participants: $M = 80.9\%$, s.d. = 8.6% correct responses and $M = 9.4\%$, s.d. = 5.6% commission errors) as well as participants whose data came with less than 10 observations in at least one of the design cells for any of the main analyses (see *Data analysis*; one participant).

We only selected trials for further analyses with a correct response in the preceding trial to control for potential effects of post-error processing (19.7% excluded). From these trials, we selected trials with an above-threshold response (i.e. baseline-corrected force of at least 0.25 a.u.) that was correct or constituted a commission error (i.e. left response to a letter assigned to the right response, or vice versa). We excluded all other erroneous trials, i.e. anticipatory above-threshold responses during fixation (less than 0.1%) as well as omissions of an above-threshold response (9.8%). Individual force profiles were baseline-corrected by subtracting the average force of the first 10 measurements after target onset from each force measurement of that response. This correction might lead to seemingly negative force values although actual negative force values cannot emerge by design. To avoid confusion, we set all force values that turned negative due to the correction procedure to zero. We then identified the maximum force (i.e. PF) of each trial, the time to PF, as well as onset and offset times.

For the same set of trials, we processed the force applied to the key that had not been pressed above-threshold and determined for each trial whether participants hit or exceeded a low-threshold force of 0.1 a.u. at least once in the trial. In correct trials, these covert responses corresponded to subthreshold commission errors and in error trials, they represented subthreshold correct responses.

We extracted forces in two ways: first, time-locked to target onset until 1600 ms after target onset (target-locked), and second, time-locked to the first occurrence of the PF in a trial, with a window from 300 ms before the peak to 300 ms after the peak (peak-locked). To allow for averaging, we employed linear interpolation to estimate force values for every millisecond in the corresponding timeframe. From the target-locked force data, we also derived the duration of each response (RD) for statistical analyses by determining the time-window between reaching a threshold of 0.25 a.u. for the first time in a trial and the time point where it reached this point for the last time in a trial. We also considered when force peaked (for the first time) in a trial, to analyse the effects of error processing on RD before and after this peak. For the peak-locked force, we computed relative force values by dividing each force value by its PF. We excluded trials from further analyses as outliers when RT, RD or PF deviated more than 2.5 standard deviations from their cell mean (4.9%).

In a next step, we matched correct and error trials by (i) their RTs and (ii) their PFs for each participant and referred to these data sources as RT-matched and PF-matched data, respectively. As error trials were less frequent than correct trials, we selected error trials one-by-one from the lowest trial number to the highest trial number. For the error trial with the lowest trial number, we subtracted the RT (PF) of that error trial from each correct trial. The correct trial with the smallest absolute difference was chosen as a match. In the case of ties, we assessed the trial number of the error trial and all tied correct trials, selecting the correct trial that lay closest to the error trial; if two trials were tied also on this latter test, we would select the trial with the smaller trial number. After each match, we proceeded to the error trial with the next higher trial number considering only correct trials without a match until every error trial had a matching correct trial.

## 2.6. Data analysis

### 2.6.1. Main analysis: response durations as a function of accuracy and time-window

Our main analysis assessed RDs in a $2 \times 2$ ANOVA with the within-subject factors accuracy (correct versus error) and time-window (pre-peak versus post-peak). We performed this analysis on the unmatched data, on the RT-matched data and on the PF-matched data, and we would only infer active error cancellation if commission errors came with shorter RDs than correct responses in all three analyses. Significant two-way interactions in the ANOVAs were followed up with separate one-tailed paired-samples $t$-tests before and after the peak to determine whether error cancellation operated pre-peak, post-peak or in both timeframes. If any of the three ANOVAs returned a non-significant effect of accuracy, we would compute Bayes factors for the pairwise comparison of RDs between correct and erroneous responses and collect additional data in increments of two participants until reaching a Bayes factor of $BF_{01} > 10$ or $BF_{01} < 0.10$ for all three analyses (using a Cauchy distribution with a scale parameter of 1 as prior) or until reaching a sample of 100 analysable datasets. We used separate Bayesian $t$-tests rather than Bayesian ANOVA, to avoid the variability of Bayes factor estimates inherent in current approaches to the latter [34].

### 2.6.2. Temporal evolution of response force

To further characterize the temporal evolution of the force profiles, we computed the AUC from the aggregated relative forces for each participant (computed on the unmatched, RT-matched and PF-matched, peak-locked data). We analysed the AUCs in $2 \times 2$ ANOVAs with the within-subject factors accuracy (correct versus error) and time-window (pre-peak versus post-peak) with follow-up tests as for RDs.

Finally, we summed AUCs of both time-windows of the unmatched data and computed differences between correct and error trials as an overall summary statistic ($\Delta_{AUC}$). One-tailed tests of the Pearson-correlations between these $\Delta_{AUC}$ and the percentage of low-threshold erroneous responses in correct trials (i.e. number of correct trials with low-threshold erroneous responses/number of correct trials) informed about the relation of error cancellation to successful abortion of subthreshold errors. In the case of a non-significant correlation, we would compute a Bayes factor using a shifted, scaled beta distribution as prior ($r$ scale parameter $= 1/3$) [35,36].

### 2.6.3. Response time and peak force analyses

To further validate our approach, we probed whether RTs and PFs were higher for correct than for erroneous responses in separate one-tailed paired-samples $t$-tests. We also checked the success of our matching procedure by employing equivalence tests [37]. In one-tailed one-sample $t$-tests, we tested whether differences between correct and erroneous responses were greater than $-2$ ms in RTs or $-0.05$ a.u. in PFs, and less than 2 ms in RTs or 0.05 a.u. in PFs. We chose these boundaries based on the differences observed in the unmatched data of our validation study, which were about 3 (RTs) or 4 (PFs) times as large as the effective boundaries (see electronic supplementary material). If differences were both significantly greater than the lower boundary and smaller than the upper boundary (we reported the test with the smaller $t$-statistic), we assumed equivalence of correct and erroneous responses. If a matching procedure did not yield comparable datasets, we would trim the distribution of the respective dependent variable (PF or RT) of the error data. We would remove the bottom 5% data points for participants whose mean difference scores were equivalent to or lower than the negative test value or equivalent to or higher than the positive test value and re-run the matching procedure. This process would be iterated until reaching a satisfactory match. If all participants were within the critical values of the equivalence tests but there was still no significant equivalence, we would trim the bottom 5% data points of erroneous responses for all participants. This process would be iterated until reaching a satisfactory match.

Finally, we determined the mean RT of the RT-matched data, which will be referred to as mean matched RT in the following. If all three AUC analyses pointed towards smaller differences between correct and erroneous responses before reaching the PF, we would conduct follow-up analyses on the time course of the force profile. More precisely, we would analyse mean forces of RT-matched data in time-windows of 20 ms preceding the mean matched RT in a $5 \times 2$ ANOVA with the within-subjects factors time-window (99 to 80 ms versus 79 to 60 ms versus 59 to 40 ms versus 39 to 20 ms versus

19 ms to mean matched RT) and accuracy (correct versus error). We would test for violations of sphericity and report Greenhouse–Geisser corrections along with the corresponding $\varepsilon$ estimate if necessary. A significant interaction of both factors would be followed up by separate one-tailed paired-samples $t$-tests to test whether correct forces were higher than erroneous forces in each time-window.

## 2.7. Validation study

We conducted a small-scale validation study to establish that the proposed study procedures, data processing routines and power considerations were feasible (see the electronic supplementary material for a detailed report; electronic supplementary material, figure S3 summarizes the corresponding results). In short, these data indicated the proposed study design and analysis plan to be feasible and the planned sample size to deliver appropriate power for all relevant effect sizes (greater than 95% for all hypotheses, greater than 99% for the main hypothesis).

# 3. Results

Figure 2 shows the main results for the unmatched data (figure 2$a$,$b$), the PF-matched data (figure 2$c$,$d$) and the RT-matched data (figure 2$e$,$f$).

## 3.1. Unmatched data

The mean force profile of erroneous responses came with a markedly smaller peak and a smaller width than the force profile of correct responses (figure 2$a$). Accordingly, RDs were shorter for error commissions than for correct responses (115 versus 145 ms; figure 2$b$; Hypothesis 1), $F_{1,33} = 127.22$, $p < 0.001$, $\eta_p^2 = 0.79$, as well as before than after reaching the peak (55 versus 75 ms), $F_{1,33} = 284.70$, $p < 0.001$, $\eta_p^2 = 0.90$. Both factors interacted (Hypothesis 2), $F_{1,33} = 15.36$, $p < 0.001$, $\eta_p^2 = 0.32$. Effects of accuracy were large before, $\Delta_{RD} = 14$ ms, $t_{33} = 11.05$, $p < 0.001$, $d_z = 1.90$, 95% CI = [11 ms, $\infty$] and after the peak, $\Delta_{RD} = 16$ ms, $t_{33} = 10.86$, $p < 0.001$, $d_z = 1.86$, 95% CI = [14 ms, $\infty$], with numerically smaller but more consistent effects before the peak.

AUCs were smaller for erroneous responses than for correct responses (155 versus 161 a.u.; figure 2$a$; Hypothesis 3), $F_{1,33} = 10.13$, $p = 0.003$, $\eta_p^2 = 0.23$, and before than after the peak (66 versus 92 a.u.), $F_{1,33} = 330.68$, $p < 0.001$, $\eta_p^2 = 0.91$. The interaction of both factors was not significant (Hypothesis 4), $F_{1,33} = 2.02$, $p = 0.164$, $\eta_p^2 = 0.06$. The correlation between $\Delta_{AUC}$ (M = 6 a.u., s.d. = 11 a.u.) and the percentage of low-threshold erroneous responses in correct trials (M = 3.11%, s.d. = 4.07%) was not significant (Hypothesis 6), $r_{34} = 0.22$, $t_{32} = 1.28$, $p = 0.104$, $BF_{10} = 1.31$.

Erroneous responses were faster than correct responses (436 versus 449 ms; Hypothesis 7), $\Delta_{RT} = 13$ ms, $t_{33} = 4.21$, $p < 0.001$, $d_z = 0.72$, 95% CI = [8 ms, $\infty$]. PFs were also smaller for erroneous responses than for correct responses (0.43 versus 0.54 a.u.; Hypothesis 8), $\Delta_{PF} = 0.11$ a.u., $t_{33} = 6.15$, $p < 0.001$, $d_z = 1.05$, 95% CI = [0.08 a.u., $\infty$].

## 3.2. Peak force-matched data

Matching led to comparable PFs for erroneous and correct responses (both 0.43 a.u.), $|t_{33}| \geq 64.44$, $p < 0.001$, $|d_z| \geq 11.05$. RDs of erroneous responses were still shorter than RDs of correct trials (115 versus 122 ms; figure 2$c$,$d$; Hypothesis 1), $F_{1,33} = 14.55$, $p = 0.001$, $\eta_p^2 = 0.31$. RDs were also shorter before than after reaching the peak (49 versus 69 ms), $F_{1,33} = 332.98$, $p < 0.001$, $\eta_p^2 = 0.91$. The interaction between both factors was significant (Hypothesis 2), $F_{1,33} = 17.54$, $p < 0.001$, $\eta_p^2 = 0.35$. RDs in erroneous trials were shorter than RDs in correct trials before the peak, $\Delta_{RD} = 2$ ms, $t_{33} = 2.30$, $p = 0.014$, $d_z = 0.40$, 95% CI = [0.48 ms, $\infty$], and this difference was larger after the peak, $\Delta_{RD} = 5$ ms, $t_{33} = 4.45$, $p < 0.001$, $d_z = 0.76$, 95% CI = [3.06 ms, $\infty$].

AUCs for PF-matched trials were smaller for erroneous than for correct responses 155 versus 161 a.u.; figure 2$c$; Hypothesis 3), $F_{1,33} = 13.50$, $p = 0.001$, $\eta_p^2 = 0.29$, and before than after the peak (66 versus 93 a.u.), $F_{1,33} = 347.76$, $p < 0.001$, $\eta_p^2 = 0.91$. Both factors entered a significant interaction (Hypothesis 4), $F_{1,33} = 16.58$, $p < 0.001$, $\eta_p^2 = 0.33$. AUCs in erroneous trials were not significantly different from AUCs in correct trials before the peak, $\Delta_{AUC} = 1$ a.u., $t_{33} = 1.43$, $p = 0.080$, $d_z = 0.25$, 95% CI = [0 a.u., $\infty$], but significantly smaller after the peak, $\Delta_{AUC} = 5$ a.u., $t_{33} = 4.57$, $p < 0.001$, $d_z = 0.78$, 95% CI = [3 a.u., $\infty$].

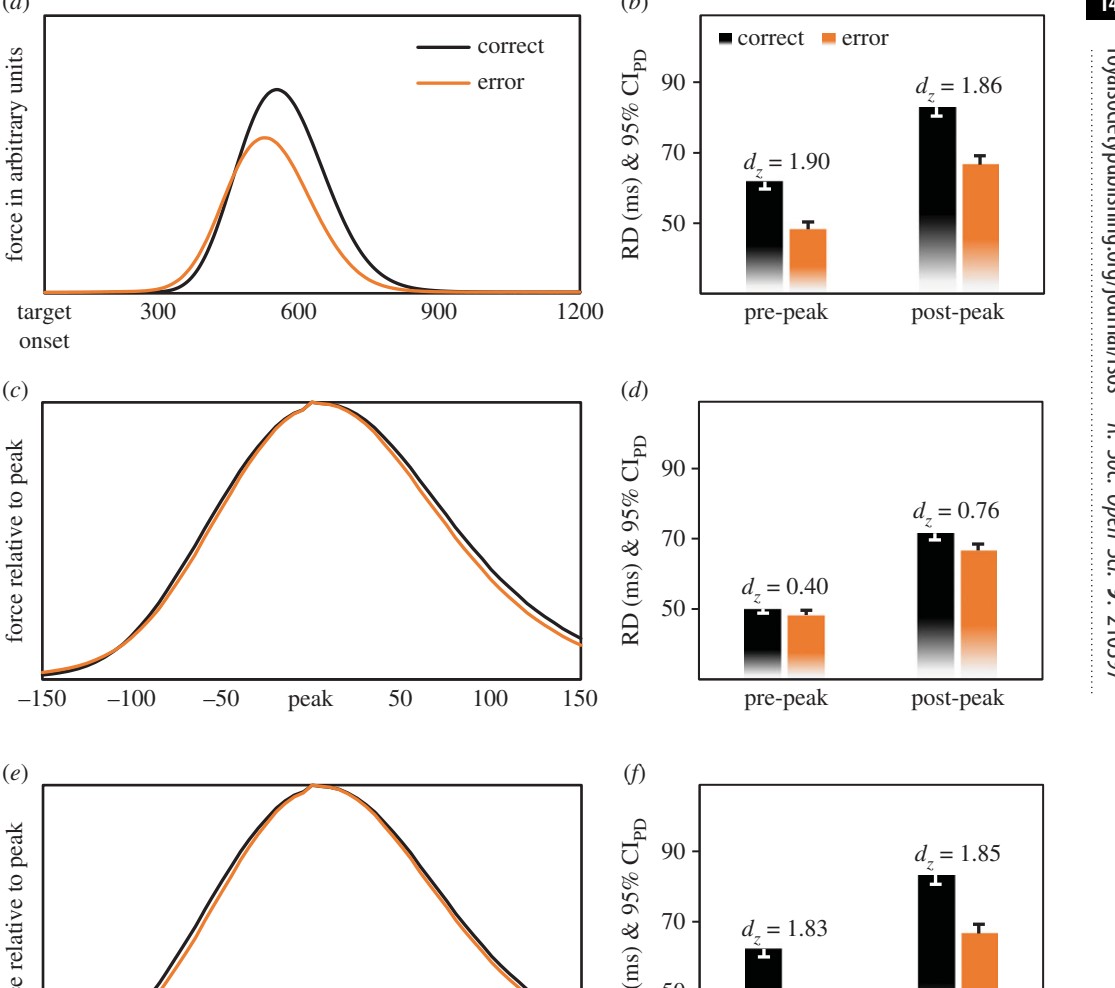

**Figure 2.** Force distribution and RD. The black lines and bars represent correct responses and the bright, orange ones represent commission errors. The upper two panels depict unmatched data as (*a*) target-locked forces in arbitrary units and (*b*) corresponding RDs. The lower four panels present data matched by PFs (*c,d*) and by RTs (*e,f*) as peak-locked forces in relative units with the peak at 100% and corresponding RDs. The plots on RD provide 95% confidence intervals of the paired differences (CI$_{PD}$) in one-tailed tests, and effect sizes $d_z$ for differences between correct and erroneous responses for the pre-peak and post-peak time-window.

## 3.3. Response time-matched data

When we matched trials by RTs, erroneous responses were not equally fast as correct responses (436 versus 437 ms), $|t_{33}| \geq 1.25$, $p \leq 0.109$, $|d_z| \geq 0.22$. The difference scores of three participants exceeded one of the test values. We therefore followed the planned follow-up steps and removed the bottom 5% of their erroneous responses (participants still provided at least 10 observations in each design cell after these exclusions) and repeated our matching procedure. This procedure provided a satisfactory match of erroneous and correct responses (both 437 ms), $|t_{33}| \geq 3.86$, $p < 0.001$, $|d_z| \geq 0.66$.

Erroneous responses had shorter RDs than correct responses (115 versus 145 ms; figure 2*e,f*; Hypothesis 1), $F_{1,33} = 122.65$, $p < 0.001$, $\eta_p^2 = 0.79$. RDs were further shorter before than after reaching the peak, (55 versus 75 ms), $F_{1,33} = 267.70$, $p < 0.001$, $\eta_p^2 = 0.89$. The two-way interaction was significant (Hypothesis 2), $F_{1,33} = 12.67$, $p = 0.001$, $\eta_P^2 = 0.28$. RDs in erroneous trials were shorter than RDs in correct trials before the peak, $\Delta_{RD} = 14$ ms, $t_{33} = 10.66$, $p < 0.001$, $d_z = 1.83$, 95% CI = [12 ms, $\infty$], and this difference increased after the peak, $\Delta_{RD} = 17$ ms $t_{33} = 10.76$, $p < 0.001$, $d_z = 1.85$, 95% CI = [14 ms, $\infty$].

AUCs for RT-matched trials were smaller for erroneous than for correct responses (155 versus 161 a.u.; Hypothesis 3), $F_{1,33} = 9.88$, $p = 0.004$, $\eta_p^2 = 0.23$, and before than after the peak (66 versus 92 a.u.), $F_{1,33} = 318.62$, $p < 0.001$, $\eta_p^2 = 0.91$. The interaction of both factors was not significant (Hypothesis 4), $F_{1,33} = 1.78$, $p = 0.191$, $\eta_p^2 = 0.05$. Hypothesis 5 was not tested because differences between correct and erroneous responses in AUCs before the peak were not significant in the PF-matched data (significant effects in all datasets were a precondition for testing Hypothesis 5).

# 4. Discussion

The current study investigated whether agents cancel erroneous actions even on the short timescale of a simple keypress. We therefore assessed RDs of keypress responses and measured the force profile of each response to scrutinize how early this process of error cancellation kicks in. Crucially, we aimed at matching correct and erroneous responses for as many surface parameters as possible—specifically: RT and PF—to arrive at a pure comparison of the corresponding duration data. Our main results indeed showed that RDs were consistently shorter for erroneous than for correct responses even when controlling for the significantly shorter RTs and lower PFs of erroneous responses (Hypothesis 1). Matching for RT did not seem to exert an impact on effects of accuracy in RD (unmatched and matched $\eta_p^2 = 0.79$). However, the size of the effect considerably dropped for the PF-matched data although it was still highly systematic ($\eta_p^2 = 0.31$). This drop might indicate, first, that error cancellation does not only reduce the duration but also the strength of execution, or second, that the same initiation and planning processes that led to reduced PFs also reduced RDs, irrespective of any error cancellation efforts. The results of the PF-matched data deliver strong support, however, that RDs reflect error cancellation proper.

Differences in RDs already emerged early during responding, namely in the time epoch that spanned from response onset to PF, and continued to be evident after the peak (Hypothesis 2). In fact, these effects were robust across datasets at both timepoints. Accordingly, error cancellation should already emerge before or during the deflection of the error-related negativity (ERN), which is at odds with the interpretation of the ERN as reflecting the earliest time point of error processing [10,11]. Early error cancellation, however, complements findings on early preparation of error-correction responses that have been reported for manual actions [17,38,39] and visual search behaviour alike [40].[3] Conflict between the execution and the cancellation of an erroneous response (here: pressing versus releasing a key) might instead contribute to the ERN. Recent data from our laboratory indeed point to a strong impact of erroneous RDs on the size of the ERN [41]. Another promising avenue to explore error detection and cancellation in this regard would be a comparison of RDs between correct and erroneous responses for subthreshold and supra-threshold responses. Early error detection during action planning might lead to traces of error cancellation for subthreshold responses. The results of the current study already demonstrate that active countermeasures against errors operate at a considerably earlier timepoint than previously assumed (see also [42,43] for evidence on early error sensations).

Erroneous responses did not only start early (Hypothesis 7), peak on a lower level (Hypothesis 8) and end earlier than correct responses, they were further enacted with less overall force than correct actions, reflected in smaller AUCs for erroneous responses in all three datasets (Hypothesis 3). Although this difference applied to both parts of the force curve, i.e. before and after the peak of the response in the unmatched and the RT-matched data, it was only evident after the peak in the PF-matched data (Hypothesis 4 and 5). Therefore, we conclude that response force is attenuated for errors especially in a late phase of responding. Together with the finding of larger differences in RDs after than before the peak, these results suggest that cancellation efforts become stronger over the course of responding erroneously, further rebutting alternative explanations of these effects in terms of response planning and initiation. The late attenuation of overall force might also explain why we did not find evidence for a relationship between this effect and successful abortion of erroneous subthreshold responses. Instead, indicators for early cancellation success, as for example particularly short RDs of erroneous responses before force peaks, might show a stronger relation with the successful cancellation of errors

---

[3]Erroneous saccades to a frequent distractor location during visual search have been shown to come with dwell times that are too small to allow for planning a new saccade only after landing on a distractor (less than 150 ms; [40]). These observations mirror early correction responses in manual tasks in that planning of a correction response starts even before the erroneous response is fully performed [38]. In contrast with manual actions, however, error correction (i.e. performing the intended correct response) and error cancellation (i.e. aborting the current erroneous response) are necessarily confounded for eye-movements. The present results suggest that low dwell times probably draw on contributions from correction and cancellation alike.

before response threshold. Further, we might have chosen a relatively weak indicator of success in the abortion of erroneous subthreshold responses. High values might indicate that agents usually prepared both responses up to a certain level whereby the correct response gained activation more rapidly in the majority of episodes without any active cancellation of the erroneous response.

The observation of immediate error cancellation also suggests that the erroneous action might be subject to inhibition. Instead, erroneous responses seem to remain even more accessible in an upcoming action episode compared with a neutral response that neither corresponded with the preceding erroneous nor the actual correct response ([25]; see also [44,45]). This observation, however, does not contradict the assumption of an overall inhibition of motor activity [19]. It would still be feasible to assume that responding is inhibited in general whereby the specific erroneous response receives somewhat less inhibition. Whether the strength of error cancellation relates to the future accessibility of the erroneous response is an open question worthy of exploration. These considerations also establish intriguing links toward theories of maladaptive and adaptive error processing [4]. Error cancellation itself qualifies as an adaptive mechanism in potentially avoiding negative consequences of the error. However, the current perspective on maladaptive and adaptive processes relates to behaviour *after* an error, error cancellation therefore calls for an extension of this perspective.

## 5. Conclusion

At a timescale that researchers have attributed to mere error detection, erroneous actions already show a reliable pattern of cancellation. We assume that error cancellation will be even more powerful for more complex actions and sequences of actions where agents still have a good chance to mitigate (some of) the consequences of their errors by cancelling ongoing motor activity.

Ethics. This research complies with the ethical regulations of the Ethics Committee of the local Institute of Psychology, the German Psychological Society and the German Research Foundation. This study qualifies for approval without individual review by the local ethics committee as participants signed informed consent, data collection is anonymous, and the task does not pose any foreseeable risk for participants.
Data accessibility. The data and analysis code for the pilot analyses and for the pre-registered main study are publicly available, as are the data and analysis code for the validation study described in the electronic supplementary material [46] and the Stage 1 protocol (osf.io/5v9es).
Authors' contributions. A.F.: conceptualization, data curation, formal analysis, investigation, methodology, project administration, software, supervision, validation, visualization, writing—original draft and writing—review and editing; M.S.: conceptualization, methodology and writing—review and editing; K.A.S.: conceptualization, methodology and writing—review and editing; W.K.: conceptualization, methodology, resources, software and writing—review and editing; R.P.: conceptualization, formal analysis, funding acquisition, methodology, project administration, resources, supervision, validation, visualization, writing—original draft and writing—review and editing.
All authors gave final approval for publication and agreed to be held accountable for the work performed therein.
Competing interests. We declare we have no competing interests.
Funding. This research was funded through a project of the German Research Foundation ('Deutsche Forschungsgemeinschaft'; PF 853/6-1) as part of the Research Unit FOR2790.

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
