## [Peer Review File · Royal Society Open Science]

Review History

Decision letter (RSOS-210358.R0)

Dear Dr Foerster,

I write you in regards to manuscript RSOS-210358 entitled "Error cancellation" which you submitted to Royal Society Open Science.

We routinely triage submissions for scientific soundness, clarity and general adherence to the Registered Reports guidelines. For submissions that have promise but are not yet suitable for in-depth Stage 1 review, we offer feedback to help authors maximise the chances that reviewers will respond positively to a resubmission.

We have concluded that your submission is not yet suitable for in-depth review and has therefore been rejected at this time, but we believe it will be suitable once several issues are addressed. We therefore invite a resubmission. Further comments from the Associate Editor may be found at the end of this letter.

If you wish to revise your manuscript in light of the below comments please submit your manuscript as a new submission and mention this previous manuscript ID in your covering letter. You should also provide a detailed response to the below comments in the cover letter.

Thank you for considering Royal Society Open Science for the publication of your registered report.

on behalf of Professor Chris Chambers (Registered Reports Editor, Royal Society Open Science)
openscience@royalsociety.org

Associate Editor Comments to Author:

The manuscript is nearly review-ready, but given the substantial number of hypotheses, to facilitate ease of understanding by reviewers please include a design table for the proposed study in the Method section of the main text based on Section 9 of this template: <https://osf.io/93znh/> You can find published examples of how these appear in these submissions (from different fields, but the principles are the same): <https://osf.io/xhdpu>, <https://osf.io/ym8gc>, <https://osf.io/g9dxb/>. Please ensure that each prediction is associated with a statistical sampling plan, a specific test (or set of tests) on specifically defined variables, and a comprehensive interpretative plan given different outcomes (i.e. a precommitment to different conclusions given different results).

Author's Response to Decision Letter for (RSOS-210358.R0)

See Appendix A.

RSOS-210397.R0

Review form: Reviewer 1

Do you have any ethical concerns with this paper?

No

Recommendation?

Major revision

Comments to the Author(s)

In this Stage 1 Registered Report, the authors proposed to investigate the active error cancellation process by looking at the force and duration of erroneous compared to correct responses. The research question is interesting and valid. The proposed hypotheses seem plausible and follow

logically from the assumed error cancellation process (although the authors may need to further elaborate on some alternative explanations and how they may rule these alternative explanations out; see my major comments below). From the pilot data and the validation study, it is also clear that both the proposed methodology and analysis pipeline are sound and feasible. The authors also provided clear and detailed information to prevent undisclosed flexibility and enable potential future replications. Lastly, the authors have considered sufficient outcome-neutral conditions, by including some validation checks and sequential sampling under the Bayesian framework in case the results are not informative. Overall, I find the proposed research quite interesting and the methodology both clear and sound. Below I list a few major and minor comments that may help the authors in further preparing this Registered Report and the proposed research.

Major comments:

- One conceptual question I have is whether reduced response duration for erroneous compared to correct responses would necessarily mean the existence of an active error cancellation process. Alternatively, errors in such simple speeded tasks could be due to lapses in attention, which in turn might lead to action slips with shortened response durations. In other words, erroneous responses may be 'born' with reduced duration, rather than it being a consequence of an active error cancellation process. I find it difficult to disentangle these two possibilities and would therefore like the authors to comment on this point. For instance, which results would show that there is indeed an active cancellation process involved when people respond incorrectly, rather than that erroneous responses are executed with shorter durations in the first place (without an active cancellation process).
- Page 5, line 26: The authors noted that erroneous responses tend to be made with smaller peak forces than correct responses, which could be an indication of active response cancellation in itself. It is not entirely clear to me then why the authors would control for PFs, if PFs may be a signature of active error cancellation. The authors may elaborate on this (e.g., what alternative explanations would remain if RTs and PFs were not controlled for) to make their reasoning clearer.
- Page 6: I find the section 'Pilot data' a bit difficult to follow on a first read, presumably because much background information is missing. The authors may provide more information on the number of participants, the number of trials, the data preparation and analysis approach etc. (e.g., it is unclear to me what the authors did exactly for the distributional analysis).
- Page 8: There appear to be some inconsistencies in the planned analysis in the power analysis (paired t tests), and the main analysis that the authors will actually use (2 by 2 ANOVAs).
- The authors may wish to provide more detailed information on the adaptive sampling procedure in case they need to add more participants. For instance, are they going to check the Bayes factor after every additional participant, or for instance after every 5 participants? Do the authors have a maximum sample size in mind where they will stop sampling even the BFs do not reach the critical values, or would the authors continue sampling (regardless of the sample size that might be required) until the BFs are either above 10 or below 1/10?

Minor comments:

- Page 4, line 55 and Page 5, line 7: Explicit definitions of both response duration and response time (i.e., how they are measured) would be helpful.
- Page 8: Would be good to cite the R packages used as well as R to give credits to R developers and the R package authors.
- Page 9, line 48: Do the target and the distractors remain on screen for 600 milliseconds, or disappear the moment participants respond?

- Page 9, line 52: "Response onsets will be identified when the force on one key is at least 0.25 arbitrary units (a.u.) above its baseline." I wonder how comparable the 0.25 a.u. threshold is to the amount of force people normally need to use to press a key on a keyboard. Can the authors provide some information on how this arbitrary unit corresponds to other units of force such as Newton or gram?
- Page 11, line 1: "Any negative force values after this correction will be set to zero." Why?
- Page 12, line 28: "BF01 > 10 or BF10 < 0.10". Should be "BF01 > 10 or BF01 < 0.1"?

Review form: Reviewer 2

Do you have any ethical concerns with this paper?

No

Recommendation?

Accept with minor revision

Comments to the Author(s)

Review Foerster et al., Error cancellation

1. The scientific validity of the research question(s).

The aim of the study is to test if error detection not only influences future behavior, but even ongoing behavior by inhibiting motor activity during an ongoing movement. This is a very appealing and novel research question. The results would be relevant for current models of error processing which encompass post-error processing affecting behavior in the next trial, but not in the current trials as proposed here. This would argue for a rapid adaptive processes of motor control that haven't been demonstrated in the context of motor error cancellation. As mentioned by the authors, such rapid error cancellation processes are especially important for real-world actions which unfold over longer time scales and different sub goals. So, I was hoping to see a more complex motor task than button presses. By saying this, I can understand the arguments for starting off with a simple motor task. It might be interesting to know if the findings on simple button presses would generalize to more complex movements or movement sequences. These may be a point for the later discussion.

The introduction sufficiently captures the current state of research and clearly points to the gap in the literature leading to the research question and a hypothesized process of active error cancellation. This is convincingly backed-up by pilot data already showing a quantifiable difference in the duration of correct and erroneous responses suggesting an online inhibition of motor errors. Regarding the theoretical concepts, it would be helpful to clarify the terms error cancellation and error attenuation. Do the authors use these terms synonymously? Cancellation is often related to an all-or-nothing process (complete absence), whereas attenuation is related to a gradual decrease. It would be helpful to define these terms and use it consistently throughout the manuscript.

Overall, the research question is clearly formulated and scientifically valid.

2. The logic, rationale, and plausibility of the proposed hypotheses.

The 8 hypotheses are logically derived from the literature and the pilot data. A specific emphasis is given to the measurement of peak forces, a suitable measure to examine the research question. Assessing error cancellation processes by studying finger forces is a novel approach that substantially extends previously used kinematic measures such as reaction time and movement duration. The introduction ends with 8 very precisely formulated hypotheses. I can follow their logic, except of one point. As described on page 5 (line28-30), 'convincing evidence for the cancellation hypothesis requires datasets that are carefully matched for RTs and PFs to allow for a

confound-free comparison of both types of responses'. Reading this, I was surprised to see that the unmatched condition is still planned to be analyzed. Moreover, exactly the same results are expected for the unmatched and matched conditions questioning the rationale of assessing both conditions. If the unmatched and matched conditions are both examined, I would have expected a comparison at some point, e.g. to justify the advantage of the latter. Further explanation is needed why the unmatched condition will still be considered given the disadvantages mentioned by the authors.

3. The soundness and feasibility of the methodology and analysis pipeline (including statistical power analysis where applicable).

The methodology and statistical analysis pipeline are described in detail and are sound. There are power calculations performed based on the pilot data and correctly described.

I have some comments that need to be addressed.

The results of the pilot study are hard to follow. This could be improved by trying to relate the description of the results more closely to the plots in figure 1. If I understand it correctly, there was an interaction effect between RT quartiles and correct/error trials. However, if I look at figure 1D such an interaction effect is not obvious. I might have misread the results or related to the wrong figure; but overall I found the presentation of the results quite confusing (same in the supplementary material).

In the experiment, participants will receive a summary on their performance with the mean RT of correct responses. I'm wondering what this absolute value can tell the participants. How does the participant know what a good/fast or poor/slow value is? Would it make more sense to feed back that they should continue as they do or that they should be faster in the next block? The same accounts for the errors. For me, it would make more sense to inform about high or low error rate instead of the exact number participants cannot evaluate as they don't know what is high and low in this specific task.

Please clarify why the first trial in each block is not analyzed. As each block contains each combination of targets and distractors twice, is then one repetition missing if the first trial is deleted or is an additional trial included as a first trial?

It is mentioned that any negative force values will be set to zero. What does a negative force value mean? How can negative values be generated and what is the rationale behind this procedure?

The data provided in the supplementary material were very convincing and ensured that the match of datasets for RTs and PFs is indeed feasible and leads to meaningful results. The authors could also apply this method to their pilot data (RT match) and see if they can achieve similar results. I'm wondering if there would be a way to actually quantify the goodness of the match for RT and PF for each participant. Difference in the goodness of the match could influence the results and therefore would be useful to report. An additional criteria based on the goodness of the match could also help to exclude participants where this procedure is not working reliably.

4. Whether the clarity and degree of methodological detail would be sufficient to replicate the proposed experimental procedures and analysis pipeline.

Some information is missing or needs to be clarified:

- Please define RT and RD in the methods of the pilot data.
- Force measurement system: What type of force sensor is used? What is the range and the linearity of the loading cell? Are the force sensors calibrated (e.g. with pre-defined weights and without weight)? The sampling rate (250 Hz) is rather low. Is this the limit of the system?
- Please add the object dimensions of the custom-built keys.
- How do the arbitrary units relate to force (N)?
- Are statistical results corrected for multiple comparisons?

5. Whether the authors provide a sufficiently clear and detailed description of the methods to prevent undisclosed flexibility in the experimental procedures or analysis pipeline.

This is the case. I have no further comments that could improve this point.

6. Whether the authors have considered sufficient outcome-neutral conditions (e.g. absence of floor or ceiling effects; positive controls; other quality checks) for ensuring that the results obtained are able to test the stated hypotheses.

This is the case. I don't see any further control condition that need to be added.

Overall, this is a scientifically strong proposal that would make a valuable contribution to the literature.

Review form: Reviewer 3

Do you have any ethical concerns with this paper?

No

Recommendation?

Accept with minor revision

Comments to the Author(s)

The study seems well designed, interesting and the paper is well written. I do have one suggestion regarding the embedding in the literature (and the clarification of writing regarding the extension the present study provides, of course fully acknowledging the value of replications). I know this might not be the main focus of the review process in this journal, nevertheless I think with a few additional clarifications the introduction of the experiment, the paper could clearly benefit. Many previous studies have used response force keys to investigate inhibiting and correcting error responses (some few examples at the end of the paper, various of these are already cited within the paper already). I think the current study could highlight some of the previous literature more, and state more explicitly what the contribution the current study aims to make. From my initial reading the contribution is the matching of RTs and PF across correct and error trials, with the addition of the distributional analyses, which I regard as very interesting.

I could not find some of the Figures referred to within the manuscript on the linked-to OSF upload.

I think the justification of sample size seems valid. I have only minor comments on the participant exclusion criterion and data selection procedures. The 60% correct criterion for inclusion seems lenient. It is a relatively simple task so performing almost 40% errors is almost guessing. I guess this is related to the strict 600 ms deadline which seems to add some speed-accuracy tradeoff. The requirement of at least 10 trials per cell seems valid for the main analyses. However, I wonder if this is a problem for the distributional analyses. For example, does this mean that some participants could have as few as 2-3 trials per bin in the distributional analysis here? If that is the case this seems somewhat problematic, how will the authors deal with this potential issue?

I have a couple of minor questions regarding the force keys. I see in the method section that the force profiles are baseline corrected using the first 10 samples. Is this from the onset of fixation? I wonder if response activation builds up in both response hands during the fixed 500ms fixation interval, and whether a pre-stimulus baseline would be more appropriate?

„After the response has been registered, the screen will be black for an inter-trial interval of 1000 ms while the force on both keys will still be measured. The offset of the response is registered

when the force matches or exceeds the threshold for the last time.“ Does the blank screen begin when the participant first exceeds the criterion, or only when they reduce the force again?

„At the end of each block, participants will receive a summary on their performance with the mean RT of correct responses and the number of commission and omission errors. They will also be urged to respond as quickly as possible while trying to avoid high numbers of errors.“ Is the between block feedback performance dependent? For example, if a VP makes too few errors/too many errors, are they asked to respond faster/slower, respectively?

„We will only select trials for further analyses with a correct response in the preceding trial to control for potential effects of post-error processing.“ I think the selection criterion of appropriate error sequences should be made more explicit. From the plots, it seems that the sequence CCEC is required, but from sentence above it seems other sequences would be included, such as ECEC and ECEE, which of course would be influenced by differences in pre-error speeding (Dudschig, C., & Jentsch, I. (2009). Speeding before and slowing after errors: Is it all just strategy?. *Brain research*, 1296, 56-62).

I wonder how the distribution of error responses across the experiment looks? For example, I would predict that more errors are committed at the beginning of the experiment as participants learn to S-R mapping (Of course, it is also conceivable that more errors are made towards the end of the experiment as tiredness/boredom sets in, and that these practice/tiredness effects could be different across participants). Related to this, how does a participant's force profile change across the duration of the experiment. From personal experience, I have often observed that the force profiles are extremely large to begin with and with experience/practice, participants become more efficient using only just enough force to trigger the response. The authors might want to take this into consideration.

The authors talk about the “successful cancellation of erroneous responses”. I wonder if there is a distinction between such successful cancellation and error correction (i.e., the subsequent execution of the correct response). I think this issue is touched upon within the proposed analyses, but I think it could be elaborated/discussed further. For example, is such error cancellation only observed on a subset of trials where the initiation of the correct response is activated? Or is this cancellation process regarded as independent of the correction process? How could this be looked into? I think this would also be a really nice aspect to analyze / present more clearly to the reader.

Roger, C., Núñez Castellar, E., Pourtois, G., & Fias, W. (2014). Changing your mind before it is too late: the electrophysiological correlates of online error correction during response selection. *Psychophysiology*, 51(8), 746-760.

Gehring, W. J., & Fencsik, D. (1999, April). Slamming on the brakes: An electrophysiological study of error response inhibition. In annual meeting of the Cognitive Neuroscience Society, Washington, DC.

Hochman, E. Y., Milman, V., & Tal, L. (2017). Evidence for aversive withdrawal response to own errors. *Acta psychologica*, 180, 147-154.

Please comment explicitly on each of the following points in your comments to the authors:

The scientific validity of the research question(s): valid

The logic, rationale, and plausibility of the proposed hypotheses: good

The soundness and feasibility of the methodology and analysis pipeline (including statistical power analysis where applicable): good

Whether the clarity and degree of methodological detail would be sufficient to replicate exactly the proposed experimental procedures and analysis pipeline: in general yes, see minor details above

Whether the authors provide a sufficiently clear and detailed description of the methods to prevent undisclosed flexibility in the experimental procedures or analysis pipeline: yes, see minor details above

Whether the authors have considered sufficient outcome-neutral conditions (e.g. positive controls) for ensuring that the results obtained are able to test the stated hypotheses: yes, taken care of

Decision letter (RSOS-210397.R0)

Dear Dr Foerster,

The Editors assigned to your Stage 1 Registered Report ("Error cancellation") have now received comments from reviewers. We would like you to revise your paper in accordance with the referee and editors suggestions which can be found below (not including confidential reports to the Editor). Please note this decision does not guarantee eventual acceptance.

Please submit a copy of your revised paper within three weeks (i.e. by the 26-May-2021). Please note that the revision deadline will expire at 00.00am on this date.

on behalf of Professor Chris Chambers (Registered Reports Editor, Royal Society Open Science)
 openscience@royalsociety.org

Associate Editor Comments to Author (Professor Chris Chambers):

Associate Editor: 1

Comments to the Author:

Three expert reviewers have now assessed the Stage 1 manuscript. The reviews are very constructive and are overall enthusiastic about the proposal. Key issues to address in revision include considering the precision of the design for isolating an active error cancellation mechanism, ensuring tight linking between the sampling plans and analysis plans, improving the clarity of reporting in the pilot results, justifying a range of design decisions and features, and providing additional details about the methods and analysis plans. In sum, although some careful work is needed in revision, the Stage 1 manuscript is within reach of in-principle acceptance; therefore a major revision invited.

Comments to Author:

Reviewer: 1

Comments to the Author(s)

In this Stage 1 Registered Report, the authors proposed to investigate the active error cancellation process by looking at the force and duration of erroneous compared to correct responses. The research question is interesting and valid. The proposed hypotheses seem plausible and follow logically from the assumed error cancellation process (although the authors may need to further elaborate on some alternative explanations and how they may rule these alternative explanations out; see my major comments below). From the pilot data and the validation study, it is also clear that both the proposed methodology and analysis pipeline are sound and feasible. The authors also provided clear and detailed information to prevent undisclosed flexibility and enable potential future replications. Lastly, the authors have considered sufficient outcome-neutral conditions, by including some validation checks and sequential sampling under the Bayesian framework in case the results are not informative. Overall, I find the proposed research quite interesting and the methodology both clear and sound. Below I list a few major and minor comments that may help the authors in further preparing this Registered Report and the proposed research.

Major comments:

- One conceptual question I have is whether reduced response duration for erroneous compared to correct responses would necessarily mean the existence of an active error cancellation process. Alternatively, errors in such simple speeded tasks could be due to lapses in attention, which in turn might lead to action slips with shortened response durations. In other words, erroneous responses may be 'born' with reduced duration, rather than it being a consequence of an active error cancellation process. I find it difficult to disentangle these two possibilities and would therefore like the authors to comment on this point. For instance, which results would show that there is indeed an active cancellation process involved when people respond incorrectly, rather than that erroneous responses are executed with shorter durations in the first place (without an active cancellation process).
- Page 5, line 26: The authors noted that erroneous responses tend to be made with smaller peak forces than correct responses, which could be an indication of active response cancellation in itself. It is not entirely clear to me then why the authors would control for PFs, if PFs may be a signature of active error cancellation. The authors may elaborate on this (e.g., what alternative explanations would remain if RTs and PFs were not controlled for) to make their reasoning clearer.

- Page 6: I find the section 'Pilot data' a bit difficult to follow on a first read, presumably because much background information is missing. The authors may provide more information on the number of participants, the number of trials, the data preparation and analysis approach etc. (e.g., it is unclear to me what the authors did exactly for the distributional analysis).
- Page 8: There appear to be some inconsistencies in the planned analysis in the power analysis (paired t tests), and the main analysis that the authors will actually use (2 by 2 ANOVAs).
- The authors may wish to provide more detailed information on the adaptive sampling procedure in case they need to add more participants. For instance, are they going to check the Bayes factor after every additional participant, or for instance after every 5 participants? Do the authors have a maximum sample size in mind where they will stop sampling even the BFs do not reach the critical values, or would the authors continue sampling (regardless of the sample size that might be required) until the BFs are either above 10 or below 1/10?

Minor comments:

- Page 4, line 55 and Page 5, line 7: Explicit definitions of both response duration and response time (i.e., how they are measured) would be helpful.
- Page 8: Would be good to cite the R packages used as well as R to give credits to R developers and the R package authors.
- Page 9, line 48: Do the target and the distractors remain on screen for 600 milliseconds, or disappear the moment participants respond?
- Page 9, line 52: "Response onsets will be identified when the force on one key is at least 0.25 arbitrary units (a.u.) above its baseline." I wonder how comparable the 0.25 a.u. threshold is to the amount of force people normally need to use to press a key on a keyboard. Can the authors provide some information on how this arbitrary unit corresponds to other units of force such as Newton or gram?
- Page 11, line 1: "Any negative force values after this correction will be set to zero." Why?
- Page 12, line 28: "BF01 > 10 or BF10 < 0.10". Should be "BF01 > 10 or BF01 < 0.1"?

Reviewer: 2

Comments to the Author(s)

Review Foerster et al., Error cancellation

1. The scientific validity of the research question(s).

The aim of the study is to test if error detection not only influences future behavior, but even ongoing behavior by inhibiting motor activity during an ongoing movement. This is a very appealing and novel research question. The results would be relevant for current models of error processing which encompass post-error processing affecting behavior in the next trial, but not in the current trials as proposed here. This would argue for a rapid adaptive processes of motor control that haven't been demonstrated in the context of motor error cancellation. As mentioned by the authors, such rapid error cancellation processes are especially important for real-world actions which unfold over longer time scales and different sub goals. So, I was hoping to see a more complex motor task than button presses. By saying this, I can understand the arguments for starting off with a simple motor task. It might be interesting to know if the findings on simple button presses would generalize to more complex movements or movement sequences. These may be a point for the later discussion.

The introduction sufficiently captures the current state of research and clearly points to the gap in the literature leading to the research question and a hypothesized process of active error cancellation. This is convincingly backed-up by pilot data already showing a quantifiable difference in the duration of correct and erroneous responses suggesting an online inhibition of motor errors. Regarding the theoretical concepts, it would be helpful to clarify the terms error cancellation and error attenuation. Do the authors use these terms synonymously? Cancellation is often related to an all-or-nothing process (complete absence), whereas attenuation is related to a

gradual decrease. It would be helpful to define these terms and use it consistently throughout the manuscript.

Overall, the research question is clearly formulated and scientifically valid.

2. The logic, rationale, and plausibility of the proposed hypotheses.

The 8 hypotheses are logically derived from the literature and the pilot data. A specific emphasis is given to the measurement of peak forces, a suitable measure to examine the research question. Assessing error cancellation processes by studying finger forces is a novel approach that substantially extends previously used kinematic measures such as reaction time and movement duration. The introduction ends with 8 very precisely formulated hypotheses. I can follow their logic, except of one point. As described on page 5 (line28-30), 'convincing evidence for the cancellation hypothesis requires datasets that are carefully matched for RTs and PFs to allow for a confound-free comparison of both types of responses'. Reading this, I was surprised to see that the unmatched condition is still planned to be analyzed. Moreover, exactly the same results are expected for the unmatched and matched conditions questioning the rationale of assessing both conditions. If the unmatched and matched conditions are both examined, I would had expected a comparison at some point, e.g. to justify the advantage of the latter. Further explanation is needed why the unmatched condition will still be considered given the disadvantages mentioned by the authors.

3. The soundness and feasibility of the methodology and analysis pipeline (including statistical power analysis where applicable).

The methodology and statistical analysis pipeline are described in detail and are sound. There are power calculations performed based on the pilot data and correctly described.

I have some comments that need to be addressed.

The results of the pilot study are hard to follow. This could be improved by trying to relate the description of the results more closely to the plots in figure 1. If I understand it correctly, there was an interaction effect between RT quartiles and correct/error trials. However, if I look at figure 1D such an interaction effect is not obvious. I might have misread the results or related to the wrong figure; but overall I found the presentation of the results quite confusing (same in the supplementary material).

In the experiment, participants will receive a summary on their performance with the mean RT of correct responses. I'm wondering what this absolute value can tell the participants. How does the participant know what a good/fast or poor/slow value is? Would it make more sense to feed back that they should continue as they do or that they should be faster in the next block? The same accounts for the errors. For me, it would make more sense to inform about high or low error rate instead of the exact number participants cannot evaluate as they don't know what is high and low in this specific task.

Please clarify why the first trial in each block is not analyzed. As each block contains each combination of targets and distractors twice, is then one repetition missing if the first trial is deleted or is an additional trial included as a first trial?

It is mentioned that any negative force values will be set to zero. What does a negative force value mean? How can negative values be generated and what is the rationale behind this procedure?

The data provided in the supplementary material were very convincing and ensured that the match of datasets for RTs and PFs is indeed feasible and leads to meaningful results. The authors could also apply this method to their pilot data (RT match) and see if they can achieve similar results. I'm wondering if there would be a way to actually quantify the goodness of the match for RT and PF for each participant. Difference in the goodness of the match could influence the results and therefore would be useful to report. An additional criteria based on the goodness of the match could also help to exclude participants where this procedure is not working reliably.

4. Whether the clarity and degree of methodological detail would be sufficient to replicate the proposed experimental procedures and analysis pipeline.

Some information is missing or need to be calcified:

- Please define RT and RD in the methods of the pilot data.
- Force measurement system: What type of force sensor is used? What is the range and the linearity of the loading cell? Are the force sensors calibrated (e.g. with pre-defined weights and without weight)? The sampling rate (250 Hz) is rather low. Is this the limit of the system?
- Please add the object dimensions of the custom-built keys.
- How do the arbitrary units relate to force (N)?
- Are statistical results corrected for multiple comparisons?

5. Whether the authors provide a sufficiently clear and detailed description of the methods to prevent undisclosed flexibility in the experimental procedures or analysis pipeline.

This is the case. I have no further comments that could improve this point.

6. Whether the authors have considered sufficient outcome-neutral conditions (e.g. absence of floor or ceiling effects; positive controls; other quality checks) for ensuring that the results obtained are able to test the stated hypotheses.

This is the case. I don't see any further control condition that need to be added.

Overall, this is a scientifically strong proposal that would make a valuable contribution to the literature.

Reviewer: 3

Comments to the Author(s)

The study seems well designed, interesting and the paper is well written. I do have one suggestion regarding the embedding in the literature (and the clarification of writing regarding the extension the present study provides, of course fully acknowledging the value of replications). I know this might not be the main focus of the review process in this journal, nevertheless I think with a few additional clarifications the introduction of the experiment, the paper could clearly benefit. Many previous studies have used response force keys to investigate inhibiting and correcting error responses (some few examples at the end of the paper, various of these are already cited within the paper already). I think the current study could highlight some of the previous literature more, and state more explicitly what the contribution the current study aims to make. From my initial reading the contribution is the matching of RTs and PF across correct and error trials, with the addition of the distributional analyses, which I regard as very interesting.

I could not find some of the Figures referred to within the manuscript on the linked-to OSF upload.

I think the justification of sample size seems valid. I have only minor comments on the participant exclusion criterion and data selection procedures. The 60% correct criterion for inclusion seems lenient. It is a relatively simple task so performing almost 40% errors is almost guessing. I guess this is related to the strict 600 ms deadline which seems to add some speed-accuracy tradeoff. The requirement of at least 10 trials per cell seems valid for the main analyses. However, I wonder if this is a problem for the distributional analyses. For example, does this mean that some participants could have as few as 2-3 trials per bin in the distributional analysis here? If that is the case this seems somewhat problematic, how will the authors deal with this potential issue?

I have a couple of minor questions regarding the force keys. I see in the method section that the force profiles are baseline corrected using the first 10 samples. Is this from the onset of fixation? I

wonder if response activation builds up in both response hands during the fixed 500ms fixation interval, and whether a pre-stimulus baseline would be more appropriate?

„After the response has been registered, the screen will be black for an inter-trial interval of 1000 ms while the force on both keys will still be measured. The offset of the response is registered when the force matches or exceeds the threshold for the last time.“ Does the blank screen begin when the participant first exceeds the criterion, or only when they reduce the force again?

„At the end of each block, participants will receive a summary on their performance with the mean RT of correct responses and the number of commission and omission errors. They will also be urged to respond as quickly as possible while trying to avoid high numbers of errors.“ Is the between block feedback performance dependent? For example, if a VP makes too few errors/too many errors, are they asked to respond faster/slower, respectively?

„We will only select trials for further analyses with a correct response in the preceding trial to control for potential effects of post-error processing.“ I think the selection criterion of appropriate error sequences should be made more explicit. From the plots, it seems that the sequence CCEC is required, but from sentence above it seems other sequences would be included, such as ECEC and ECEE, which of course would be influenced by differences in pre-error speeding (Dudschig, C., & Jentzsch, I. (2009). Speeding before and slowing after errors: Is it all just strategy?. *Brain research*, 1296, 56-62).

I wonder how the distribution of error responses across the experiment looks? For example, I would predict that more errors are committed at the beginning of the experiment as participants learn to S-R mapping (Of course, it is also conceivable that more errors are made towards the end of the experiment as tiredness/boredom sets in, and that these practice/tiredness effects could be different across participants). Related to this, how does a participant's force profile change across the duration of the experiment. From personal experience, I have often observed that the force profiles are extremely large to begin with and with experience/practice, participants become more efficient using only just enough force to trigger the response. The authors might want to take this into consideration.

The authors talk about the “successful cancelation of erroneous responses”. I wonder if there is a distinction between such successful cancelation and error correction (i.e., the subsequent execution of the correct response). I think this issue is touched upon within the proposed analyses, but I think it could be elaborated/discussed further. For example, is such error cancelation only observed on a subset of trials where the initiation of the correct response is activated? Or is this cancellation process regarded as independent of the correction process? How could this be looked into? I think this would also be a really nice aspect to analyze / present more clearly to the reader.

Roger, C., Núñez Castellar, E., Pourtois, G., & Fias, W. (2014). Changing your mind before it is too late: the electrophysiological correlates of online error correction during response selection. *Psychophysiology*, 51(8), 746-760.

Gehring, W. J., & Fencsik, D. (1999, April). Slamming on the brakes: An electrophysiological study of error response inhibition. In annual meeting of the Cognitive Neuroscience Society, Washington, DC.

Hochman, E. Y., Milman, V., & Tal, L. (2017). Evidence for aversive withdrawal response to own errors. *Acta psychologica*, 180, 147-154.

Please comment explicitly on each of the following points in your comments to the authors:

The scientific validity of the research question(s): valid

The logic, rationale, and plausibility of the proposed hypotheses: good

The soundness and feasibility of the methodology and analysis pipeline (including statistical power analysis where applicable): good

Whether the clarity and degree of methodological detail would be sufficient to replicate exactly the proposed experimental procedures and analysis pipeline: in general yes, see minor details above

Whether the authors provide a sufficiently clear and detailed description of the methods to prevent undisclosed flexibility in the experimental procedures or analysis pipeline: yes, see minor details above

Whether the authors have considered sufficient outcome-neutral conditions (e.g. positive controls) for ensuring that the results obtained are able to test the stated hypotheses: yes, taken care of

Author's Response to Decision Letter for (RSOS-210397.R0)

See Appendix B.

RSOS-210397.R1

Review form: Reviewer 1

Do you have any ethical concerns with this paper?

No

Recommendation?

Accept in principle

Comments to the Author(s)

The authors have satisfactorily addressed all of my comments. I look forward to seeing the results of this project.

Review form: Reviewer 2

Do you have any ethical concerns with this paper?

No

Recommendation?

Accept in principle

Comments to the Author(s)

The authors successfully answered my questions and accordingly revised the manuscripts. I have no further points to add and I'm looking forward to seeing the results.

Review form: Reviewer 3**Do you have any ethical concerns with this paper?**

No

Recommendation?

Accept in principle

Comments to the Author(s)

The authors did take care of all my issues I had with the previous version of the manuscript. I think this is a really nice study planned, and I'm looking forward to the outcome.

Decision letter (RSOS-210397.R1)

Dear Dr Foerster

On behalf of the Editor, I am pleased to inform you that your Manuscript RSOS-210397.R1 entitled "Error cancellation" has been accepted in principle for publication in Royal Society Open Science. The reviewers' and editors' comments are included at the end of this email.

You may now progress to Stage 2 and complete the study as approved. Before commencing data collection we ask that you:

- 1) Update the journal office as to the anticipated completion date of your study.
- 2) Register your approved protocol on the Open Science Framework (<https://osf.io/>) or other recognised repository, either publicly or privately under embargo until submission of the Stage 2 manuscript. Please note that a time-stamped, independent registration of the protocol is mandatory under journal policy, and manuscripts that do not conform to this requirement cannot be considered at Stage 2. The protocol should be registered unchanged from its current approved state, with the time-stamp preceding implementation of the approved study design. We strongly recommend using the dedicated registration portal for Stage 1 RRs at <https://osf.io/rr>

Following completion of your study, we invite you to resubmit your paper for peer review as a Stage 2 Registered Report. Please note that your manuscript can still be rejected for publication at Stage 2 if the Editors consider any of the following conditions to be met:

- The results were unable to test the authors' proposed hypotheses by failing to meet the approved outcome-neutral criteria.
- The authors altered the Introduction, rationale, or hypotheses, as approved in the Stage 1 submission.

- The authors failed to adhere closely to the registered experimental procedures. Please note that any deviations from the approved experimental procedures must be communicated to the editor immediately for approval, and prior to the completion of data collection. Failure to do so can result in revocation of in-principle acceptance and rejection at Stage 2 (see complete guidelines for further information).
- Any post-hoc (unregistered) analyses were either unjustified, insufficiently caveated, or overly dominant in shaping the authors' conclusions.
- The authors' conclusions were not justified given the data obtained.

We encourage you to read the complete guidelines for authors concerning Stage 2 submissions at <https://royalsocietypublishing.org/rsos/registered-reports#ReviewerGuideRegRep>. Please especially note the requirements for data sharing, reporting the URL of the independently registered protocol, and that withdrawing your manuscript will result in publication of a Withdrawn Registration.

Once again, thank you for submitting your manuscript to Royal Society Open Science and we look forward to receiving your Stage 2 submission. If you have any questions at all, please do not hesitate to get in touch. We look forward to hearing from you shortly with the anticipated submission date for your stage two manuscript.

on behalf of Professor Chris Chambers (Registered Reports Editor, Royal Society Open Science)
openscience@royalsociety.org

Associate Editor Comments to Author (Professor Chris Chambers):

Associate Editor: 1

Comments to the Author:

All reviewers are now satisfied and IPA can be awarded.

Reviewers' comments to Author:

Reviewer: 1

Comments to the Author(s)

The authors have satisfactorily addressed all of my comments. I look forward to seeing the results of this project.

Reviewer: 2

Comments to the Author(s)

The authors successfully answered my questions and accordingly revised the manuscripts. I have no further points to add and I'm looking forward to seeing the results.

Reviewer: 3

Comments to the Author(s)

The authors did take care of all my issues I had with the previous version of the manuscript. I think this is a really nice study planned, and I'm looking forward to the outcome.

Author's Response to Decision Letter for (RSOS-210397.R1)

See Appendix C.

RSOS-210397.R2

Review form: Reviewer 1

Is the manuscript scientifically sound in its present form?

Yes

Are the interpretations and conclusions justified by the results?

Yes

Is the language acceptable?

Yes

Do you have any ethical concerns with this paper?

No

Recommendation?

Accept with minor revision

Comments to the Author(s)

This is a Stage 2 report of a previously approved study protocol. The authors found an early error cancellation process, as they initially predicted. The introduction, rationale and hypotheses are the same as the Stage 1 report, except for some minor textual changes (e.g., changes in the tenses). The authors also closely followed the registered procedure, with one change transparently reported. The matching procedures were successful, thus the data are able to address the stated hypotheses. The conclusions are also justified given the data. Overall, I find this Stage 2 report a solid, rigorous and interesting piece of work. I have only a few minor comments, and I hope they will be helpful in preparing the paper for publication.

The raw data files are shared publicly on OSF. However, it would be helpful to future researchers if the authors could provide a codebook, explaining the meaning of the variables in the data files.

Abstract: Now that the authors have conducted the planned experiment, it would be good to also include the main findings in the abstract, rather than only the hypothesis.

Figure 2 shows the results based on the unmatched data and the PF-matched data. Since the authors also conducted analyses on the RT-matched data, I think it would be informative to show those results in the figure as well.

Page 16, line 53: The text says AUCs but the subscript is PF.

Figure 3: I find Figure 3 a bit difficult to understand, maybe because it shows the difference between two change scores over time. More importantly, it's unclear to me what information Figure 3 conveys. The authors mentioned that the gradients for the force profile were plotted, but this figure was never discussed in the text.

Review form: Reviewer 2

Is the manuscript scientifically sound in its present form?

Yes

Are the interpretations and conclusions justified by the results?

Yes

Is the language acceptable?

Yes

Do you have any ethical concerns with this paper?

No

Recommendation?

Accept with minor revision

Comments to the Author(s)

In this study, Foerster and colleagues showed that active error cancellation occurs much earlier than previous studies suggested. They followed a new approach by examining force profiles of button press responses and took great care in matching the correct and erroneous responses by response time and peak forces to provide a valid comparison of the two response types. The results of the main experiment match the ones of the validation study and confirm the pre-defined hypotheses. One of the main results is that erroneous responses led to shorter response durations within the first 100ms after movement onset. Based on the clear results, the discussion is straight-forward and concise and conveys a clear message.

Overall, this is a very convincing study. I only have some small recommendations to improve the result section and the figures.

Results: The authors described the hypotheses in very great detail at the of the introduction. In total, they hypothesized eight different outcomes. As I've read the result section, I missed a link between the hypotheses and the single results reported there. I suggest to add the hypotheses to the results to provide the reader a better guidance.

Figure 2B, 2D: The values on the y-axis are floating in the air. I would add ticks to indicate where exactly the values are located along the axis (same for Figure S3). In addition, the error bars are hard to see. I suggest to use full instead of half error bars (also to make it consistent across the bars).

Figure S3B, S3D: Please add error bars.

Page 5, typo: RDs spreads -> RD spread

Decision letter (RSOS-210397.R2)

Dear Dr Foerster:

On behalf of the Editor, I am pleased to inform you that your Stage 2 Registered Report RSOS-210397.R2 entitled "Error cancellation" has been deemed suitable for publication in Royal Society Open Science subject to minor revision in accordance with the referee suggestions. Please find the referees' comments at the end of this email.

The reviewers and Subject Editor have recommended publication, but also suggest some minor revisions to your manuscript. We invite you to respond to the comments and revise your manuscript. Below the referees' and Editors' comments (where applicable) we provide additional

requirements. Final acceptance of your manuscript is dependent on these requirements being met. We provide guidance below to help you prepare your revision.

Please submit your revised manuscript and required files (see below) no later than 7 days from today's (ie 21-Jan-2022) date. Note: the ScholarOne system will 'lock' if submission of the revision is attempted 7 or more days after the deadline. If you do not think you will be able to meet this deadline please contact the editorial office immediately.

on behalf of Professor Chris Chambers
(Registered Reports Editor, Royal Society Open Science)
openscience@royalsociety.org

Associate Editor Comments to Author (Professor Chris Chambers):

Associate Editor: 1

Comments to the Author:

Two of the original reviewers who assessed the Stage 1 submission kindly returned to evaluate the Stage 2 manuscript. As you will see the reviews are very positive, and from my own reading this is clearly a well conducted study and a fine example of a completed RR. The reviewers do offer some minor suggestions for improving clarity in the presentation of results (and OSF archive) which I think are sensible and well worth implementing. Concerning the Abstract, please do include an overview of the results and conclusions as suggested by Reviewer 1. Provided you are able to respond comprehensively to these points in a revised manuscript, final acceptance should be forthcoming without requiring further in-depth review.

Comments to Author:

Reviewer: 1

Comments to the Author(s)

This is a Stage 2 report of a previously approved study protocol. The authors found an early error cancellation process, as they initially predicted. The introduction, rationale and hypotheses are the same as the Stage 1 report, except for some minor textual changes (e.g., changes in the tenses). The authors also closely followed the registered procedure, with one change transparently reported. The matching procedures were successful, thus the data are able to address the stated hypotheses. The conclusions are also justified given the data. Overall, I find this Stage 2 report a solid, rigorous and interesting piece of work. I have only a few minor comments, and I hope they will be helpful in preparing the paper for publication.

The raw data files are shared publicly on OSF. However, it would be helpful to future researchers if the authors could provide a codebook, explaining the meaning of the variables in the data files.

Abstract: Now that the authors have conducted the planned experiment, it would be good to also include the main findings in the abstract, rather than only the hypothesis.

Figure 2 shows the results based on the unmatched data and the PF-matched data. Since the authors also conducted analyses on the RT-matched data, I think it would be informative to show those results in the figure as well.

Page 16, line 53: The text says AUCs but the subscript is PF.

Figure 3: I find Figure 3 a bit difficult to understand, maybe because it shows the difference between two change scores over time. More importantly, it's unclear to me what information Figure 3 conveys. The authors mentioned that the gradients for the force profile were plotted, but this figure was never discussed in the text.

Reviewer: 2

Comments to the Author(s)

In this study, Foerster and colleagues showed that active error cancellation occurs much earlier than previous studies suggested. They followed a new approach by examining force profiles of button press responses and took great care in matching the correct and erroneous responses by response time and peak forces to provide a valid comparison of the two response types. The results of the main experiment match the ones of the validation study and confirm the pre-defined hypotheses. One of the main results is that erroneous responses led to shorter response durations within the first 100ms after movement onset. Based on the clear results, the discussion is straight-forward and concise and conveys a clear message.

Overall, this is a very convincing study. I only have some small recommendations to improve the result section and the figures.

Results: The authors described the hypotheses in very great detail at the of the introduction. In total, they hypothesized eight different outcomes. As I've read the result section, I missed a link between the hypotheses and the single results reported there. I suggest to add the hypotheses to the results to provide the reader a better guidance.

Figure 2B, 2D: The values on the y-axis are floating in the air. I would add ticks to indicate where exactly the values are located along the axis (same for Figure S3). In addition, the error bars are hard to see. I suggest to use full instead of half error bars (also to make it consistent across the bars).

Figure S3B, S3D: Please add error bars.

Page 5, typo: RDs spreads -> RD spread

===PREPARING YOUR MANUSCRIPT===

one version should clearly identify all the changes that have been made (for instance, in coloured highlight, in bold text, or tracked changes);

===PREPARING YOUR REVISION IN SCHOLARONE===

- If you are providing image files for potential cover images, please upload these at this step, and inform the editorial office you have done so. You must hold the copyright to any image provided.
- A copy of your point-by-point response to referees and Editors. This will expedite the preparation of your proof.

- Ensure that your data access statement meets the requirements at <https://royalsociety.org/journals/authors/author-guidelines/#data>. You should ensure that you cite the dataset in your reference list. If you have deposited data etc in the Dryad repository, please only include the 'For publication' link at this stage. You should remove the 'For review' link.
- If you are requesting an article processing charge waiver, you must select the relevant waiver option (if requesting a discretionary waiver, the form should have been uploaded, see 'File upload' above).
- If you have uploaded any electronic supplementary (ESM) files, please ensure you follow the guidance at <https://royalsociety.org/journals/authors/author-guidelines/#supplementary-material> to include a suitable title and informative caption. An example of appropriate titling and captioning may be found at https://figshare.com/articles/Table_S2_from_Is_there_a_trade-off_between_peak_performance_and_performance_breadth_across_temperatures_for_aerobic_scope_in_teleost_fishes_/3843624.

Author's Response to Decision Letter for (RSOS-210397.R2)

See Appendix D.

Decision letter (RSOS-210397.R3)

Dear Dr Foerster:

It is a pleasure to accept your Stage 2 Registered Report entitled "Error cancellation" in its current form for publication in Royal Society Open Science.

Thank you for your fine contribution. On behalf of the Editors of Royal Society Open Science, we look forward to your continued contributions to the journal.

on behalf of Chris Chambers (Subject Editor)
openscience@royalsociety.org

Appendix A

Error cancellation: Cover letter

Dr. Anna Foerster

Department for Cognitive Psychology, University of Würzburg

Röntgenring 11, 97070 Würzburg, Germany

Tel. +49 931 318901, Email: anna.foerster@uni-wuerzburg.de

Resubmission of RSOS-210358 – *Error cancellation*

Dear Dr. Chambers,

Thank you very much for the positive comments on our manuscript. Please find our response below.

The manuscript is nearly review-ready, but given the substantial number of hypotheses, to facilitate ease of understanding by reviewers please include a design table for the proposed study in the Method section of the main text based on Section 9 of this template: <https://osf.io/93znh/> You can find published examples of how these appear in these submissions (from different fields, but the principles are the same): <https://osf.io/xhdpu>, <https://osf.io/ym8gc>, <https://osf.io/g9dxb/>. Please ensure that each prediction is associated with a statistical sampling plan, a specific test (or set of tests) on specifically defined variables, and a comprehensive interpretative plan given different outcomes (i.e. a precommitment to different conclusions given different results).

We have added a detailed design table to the end of the manuscript, including a sampling plan (power analysis), specific tests and interpretation given different outcomes for each hypothesis.

As stated in the original cover letter, we will register our approved protocol on the *Open Science Framework* in case of an in principle acceptance at *Royal Society Open Science*. If we were to withdraw the paper, we agree that *Royal Society Open Science* publishes a short summary of the pre-registered study under the section *Withdrawn Registrations*.

With best regards,

Anna Foerster

On behalf of the authors

Appendix B

Error cancellation: Response to reviewers

Reviewer 1

In this Stage 1 Registered Report, the authors proposed to investigate the active error cancellation process by looking at the force and duration of erroneous compared to correct responses. The research question is interesting and valid. The proposed hypotheses seem plausible and follow logically from the assumed error cancellation process (although the authors may need to further elaborate on some alternative explanations and how they may rule these alternative explanations out; see my major comments below). From the pilot data and the validation study, it is also clear that both the proposed methodology and analysis pipeline are sound and feasible. The authors also provided clear and detailed information to prevent undisclosed flexibility and enable potential future replications. Lastly, the authors have considered sufficient outcome-neutral conditions, by including some validation checks and sequential sampling under the Bayesian framework in case the results are not informative. Overall, I find the proposed research quite interesting and the methodology both clear and sound. Below I list a few major and minor comments that may help the authors in further preparing this Registered Report and the proposed research.

Thank you very much for your constructive feedback on our manuscript! Below we provide detailed responses to each of your comments.

- One conceptual question I have is whether reduced response duration for erroneous compared to correct responses would necessarily mean the existence of an active error cancellation process. Alternatively, errors in such simple speeded tasks could be due to lapses in attention, which in turn might lead to action slips with shortened response durations. In other words, erroneous responses may be 'born' with reduced duration, rather than it being a consequence of an active error cancellation process. I find it difficult to disentangle these two possibilities and would therefore like the authors to comment on this point. For instance, which results would show that there is indeed an active cancellation process involved when people respond incorrectly, rather than that erroneous responses are executed with shorter durations in the first place (without an active cancellation process).

This is an important point, and we agree that additional detail on our reasoning is warranted here. Actually, this line of thought is the main reason why we focus so heavily on different matching procedures. If we simply observed shorter durations and smaller peak forces for errors than for correct responses (as previous studies did), we would have limited grounds to argue for active cancellation as compared to errors simply being generated (aka 'born') differently. In a relatively ballistic response such as a simple keypress, we would argue that most responses should follow a canonical force profile that is mainly dependent on how forceful the movement is supposed to be. Observing differences in how the response force evolves for movements with matched peaks, however, cannot be easily explained by differences in how responses are generated. We therefore take shorter durations even for the matched data sets as evidence for active cancellation.

At the same time, it would be of course possible to come up with an elaborate (albeit likely twisted) argument how errors could come with an entirely different force profile even after matching for response times and especially for peak force. An interpretation in terms of active cancellation seems to be much more parsimonious and thus favourable, however. Perhaps needless to add, that it was the a priori assumption behind this project, whereas other explanations appear to come after the fact. We also agree that we could have stated more explicitly that any interpretation of active cancellation vs. differences in generation will have to rely on an argument of plausibility and we have done so in the revision.

• Page 5, line 26: The authors noted that erroneous responses tend to be made with smaller peak forces than correct responses, which could be an indication of active response cancellation in itself. It is not entirely clear to me then why the authors would control for PFs, if PFs may be a signature of active error cancellation. The authors may elaborate on this (e.g., what alternative explanations would remain if RTs and PFs were not controlled for) to make their reasoning clearer.

Please see our response to the above point. In a nutshell, it is this matching procedure that provides evidence either for or against active cancellation. It further allows speculations about early versus late cancellation by observing systematic changes in the force profile either before reaching the peak or only afterwards. We now explain this more explicitly in the revision.

You are also correct in pointing out that smaller PFs for errors as compared to correct responses might themselves be an indication for active error cancellation, however such observations can always be explained equally easily in terms of peculiarities of how erroneous responses are generated. That is: Errors might be initiated in a way that aims at a smaller PF from the get-go, or they might become attenuated on the fly. A convincing argument for the latter therefore requires PF-matched data.

• Page 6: I find the section 'Pilot data' a bit difficult to follow on a first read, presumably because much background information is missing. The authors may provide more information on the number of participants, the number of trials, the data preparation and analysis approach etc. (e.g., it is unclear to me what the authors did exactly for the distributional analysis).

We had originally aimed to keep this section as succinct as possible. We see that we went too far with this strategy and included the requested information to improve accessibility for the reader.

• Page 8: There appear to be some inconsistencies in the planned analysis in the power analysis (paired t tests), and the main analysis that the authors will actually use (2 by 2 ANOVAs).

The 2×2 ANOVAs have only factors with two levels each. In this case, there is a direct relation between the ANOVA and paired t-tests with $F = t^2$. This relation holds true for both main effects and the interaction alike. As such $d_z (= \frac{t}{\sqrt{n}} = \frac{\sqrt{F}}{\sqrt{n}})$ can be considered for the power analysis of the ANOVA, and we prefer this metric over f^2 , η^2 or ω^2 because Cohen's d is much more graspable in our reading. Our original phrasing in the *Sample* section might have been misleading: "...to detect the effect size corresponding to this lower bound in a two-tailed paired t-test...". So we changed it to: "...to detect the effect size corresponding to this lower bound in a two-tailed test...".

• The authors may wish to provide more detailed information on the adaptive sampling procedure in case they need to add more participants. For instance, are they going to check the Bayes factor after every additional participant, or for instance after every 5 participants? Do the authors have a maximum sample size in mind where they will stop sampling even the BFs do not reach the critical values, or would the authors continue sampling (regardless of the sample size that might be required) until the BFs are either above 10 or below 1/10?

This is very helpful advice, thank you. We will collect additional data and analyze the data in the smallest increment that allows for a counterbalanced design, that is two participants. We will stop data collection upon having 100 analyzable datasets because this sample size would be about five times bigger than the required sample size of the power calculation for the main analysis. We added this information to the first paragraph of *Data analysis*.

- Page 4, line 55 and Page 5, line 7: Explicit definitions of both response duration and response time (i.e., how they are measured) would be helpful.

We added definitions as suggested.

- Page 8: Would be good to cite the R packages used as well as R to give credits to R developers and the R package authors.

Done.

- Page 9, line 48: Do the target and the distractors remain on screen for 600 milliseconds, or disappear the moment participants respond?

Both disappear the moment participants respond. We clarified that in the text.

- Page 9, line 52: “Response onsets will be identified when the force on one key is at least 0.25 arbitrary units (a.u.) above its baseline.” I wonder how comparable the 0.25 a.u. threshold is to the amount of force people normally need to use to press a key on a keyboard. Can the authors provide some information on how this arbitrary unit corresponds to other units of force such as Newton or gram?

This threshold of 0.25 a.u. is about 250 g or 2.5 Newton. As we state in the text, this threshold yielded a reasonable amount of omission errors with this response criterion while ensuring that the keys operate with sufficient sensitivity. Further, none of the participants commented positively or negatively on the force they had to apply to respond so that we are confident that the apparatus mirrors common keyboard devices. This information has been added to the manuscript.

We also discussed converting these arbitrary units to other measures such as Newton or gram. However, because we cannot guarantee that the conversion is fully independent of peripheral factors such as room temperature, we ultimately chose to retain arbitrary units for the sake of utmost accuracy. Errors would be marginal but we would still feel more comfortable with reporting the data in arbitrary units to report the results as soundly as possible.

- Page 11, line 1: “Any negative force values after this correction will be set to zero.” Why?

By design there cannot be negative force values when operating the keys (the frame of the key does not allow for lifting the moving part of the key so that the only possible force vector is towards the table surface). These negative values only emerge because we baseline-correct force measurements by subtracting the first ten force measurements of each trial. We believe that negative force in this context might be unintuitive to many readers and therefore decided to set any negative values to zero (note that this procedural choice did not affect the pattern of results for all validation analyses). We added this reasoning to the text.

- Page 12, line 28: “BF01 > 10 or BF10 < 0.10”. Should be “BF01 > 10 or BF01 < 0.1”?

Thanks for pointing us to this typo.

Reviewer 2

1. The scientific validity of the research question(s).

The aim of the study is to test if error detection not only influences future behavior, but even ongoing behavior by inhibiting motor activity during an ongoing movement. This is a very appealing and novel research question. The results would be relevant for current models of error processing which encompass post-error processing affecting behavior in the next trial, but not in the current trials as proposed here. This would argue for a rapid adaptive processes of motor control that haven't been demonstrated in the context of motor error cancellation. As mentioned by the authors, such rapid error cancellation processes are especially important for real-world actions which unfold over longer time scales and different sub goals. So, I was hoping to see a more complex motor task than button presses. By saying this, I can understand the arguments for starting off with a simple motor task. It might be interesting to know if the findings on simple button presses would generalize to more complex movements or movement sequences. These may be a point for the later discussion. The introduction sufficiently captures the current state of research and clearly points to the gap in the literature leading to the research question and a hypothesized process of active error cancellation. This is convincingly backed-up by pilot data already showing a quantifiable difference in the duration of correct and erroneous responses suggesting an online inhibition of motor errors. Regarding the theoretical concepts, it would be helpful to clarify the terms error cancellation and error attenuation. Do the authors use these terms synonymously? Cancellation is often related to an all-or-nothing process (complete absence), whereas attenuation is related to a gradual decrease. It would be helpful to define these terms and use it consistently throughout the manuscript. Overall, the research question is clearly formulated and scientifically valid.

Thank you for these kind words about our manuscript and your nuanced feedback! We will make sure to discuss our results for more complex actions, for which error cancellation is certainly as interesting as for the studied ballistic movements.

Cancellation vs. attenuation: In the manuscript, we describe errors as being attenuated mostly when referring to the distribution of the force profile (PF and AUC) rather than the duration of the force application. We would argue that attenuation can be a precursor of response cancellation but at the same time, attenuation of force might also prolong (weak) responding, which would not qualify as response cancellation. We revised the abstract and the introduction to clarify this distinction and think that this made our argument even stronger, thank you.

2. The logic, rationale, and plausibility of the proposed hypotheses.

The 8 hypotheses are logically derived from the literature and the pilot data. A specific emphasis is given to the measurement of peak forces, a suitable measure to examine the research question. Assessing error cancellation processes by studying finger forces is a novel approach that substantially extends previously used kinematic measures such as reaction time and movement duration. The introduction ends with 8 very precisely formulated hypotheses. I can follow their logic, except of one point. As described on page 5 (line28-30), 'convincing evidence for the cancellation hypothesis requires datasets that are carefully matched for RTs and PFs to allow for a confound-free comparison of both types of responses'. Reading this, I was surprised to see that the unmatched condition is still planned to be analyzed. Moreover, exactly the same results are expected for the unmatched and matched conditions questioning the rationale of assessing both conditions. If the unmatched and matched conditions are both examined, I would had expected a comparison at some point, e.g. to justify the advantage of the latter. Further explanation is needed why the unmatched condition will still be considered given the disadvantages mentioned by the authors

We are happy that our hypotheses are overall accessible and convincing. We think that it is helpful to include the results of the unmatched data especially for comparison with previous results – including the reported pilot analyses. Crucially, as we now explain in more detail in the revised introduction (see also the first two points of Reviewer 1), RDs could be smaller for erroneous than correct responses simply because of smaller PFs while this difference might completely vanish after matching. Such a possibility can be assessed by comparing the unmatched and matched results. Moreover, Reviewer 3 pointed us to a recent study that we had overlooked and that showed reduced RDs for erroneous responses, however, without controlling for differences in RTs or PFs. As such, reporting results of the unmatched data allows for a conceptual replication of this study. Regarding the statistical comparison of accuracy effects on RDs in the different dataset, we will resort to descriptive comparison of effect size considering that the matched datasets include the same but just fewer datapoints than the unmatched data. Yet, you are of course correct that the main results of interest relate to the matched data, and we clarify this explicitly in the revision.

3. The soundness and feasibility of the methodology and analysis pipeline (including statistical power analysis where applicable).

The methodology and statistical analysis pipeline are described in detail and are sound. There are power calculations performed based on the pilot data and correctly described.

I have some comments that need to be addressed.

The results of the pilot study are hard to follow. This could be improved by trying to relate the description of the results more closely to the plots in figure 1. If I understand it correctly, there was an interaction effect between RT quartiles and correct/error trials. However, if I look at figure 1D such an interaction effect is not obvious. I might have misread the results or related to the wrong figure; but overall I found the presentation of the results quite confusing (same in the supplementary material).

We improved the accessibility of the pilot analyses by adding details about data treatment and analyses based on the comments of another reviewer. It is indeed tricky to make out the interaction effect in Figure 1D because of the large differences in RTs between RT-Quartiles (this is a common issue in percentile plots). Further, the effect of accuracy on RTs is significant for each RT-Quartile and only increases moderately. We reduced the span of the values on the y-axis as much as possible to make this increase more visible. We also included references to panels C and D of Figure 1 in the text, which we seem to have missed to do before.

In the experiment, participants will receive a summary on their performance with the mean RT of correct responses. I'm wondering what this absolute value can tell the participants. How does the participant know what a good/fast or poor/slow value is? Would it make more sense to feed back that they should continue as they do or that they should be faster in the next block? The same accounts for the errors. For me, it would make more sense to inform about high or low error rate instead of the exact number participants cannot evaluate as they don't know what is high and low in this specific task.

We agree that absolute response times and errors might not be the most accessible information for participants initially. After the first block, however, they have their previous feedback as a reference value, and many participants are actually quite motivated to improve their performance throughout the experiment (based on experience in previous studies). We therefore prefer this type of feedback over general reminders such as “try to respond more quickly” or “respond more accurately”, because it might be frustrating for generally poor-performing participants or hardly motivating for generally excellent-performing participants. Given the success of our validation study in the observation of a sufficient number of observations, we think it is reasonable to keep this procedure, also in keeping with standard procedures at our institute (which many participants are accustomed to).

Please clarify why the first trial in each block is not analyzed. As each block contains each combination of targets and distractors twice, is then one repetition missing if the first trial is deleted or is an additional trial included as a first trial?

We exclude the first trial of each block for two reasons. For one, performance on the first trial after a break tends to be somewhat noisy (commonly referred to as “restart costs”), as participants have to refocus on the task. Second, we only analyze trials that follow a correct trial to control for effects of post-error processing. The first trial of each block has no preceding trial and is therefore different from these post-correct trials. We do not replace this first trial with its specific stimuli because the randomization procedure should preclude any systematic biases.

It is mentioned that any negative force values will be set to zero. What does a negative force value mean? How can negative values be generated and what is the rationale behind this procedure?

By design there cannot be negative force values when operating the keys. These negative values only emerge because we baseline-correct force measurements by subtracting the first ten force measurements (measured before presentation of the target letter) of each trial. So if participants reduce force below the mean force of the first ten measurements, there will be negative values. We figured that the data would be more accessible if any negative values are set to zero. We added this reasoning to the text.

The data provided in the supplementary material were very convincing and ensured that the match of datasets for RTs and PFs is indeed feasible and leads to meaningful results. The authors could also apply this method to their pilot data (RT match) and see if they can achieve similar results. I'm wondering if there would be a way to actually quantify the goodness of the match for RT and PF for each participant. Difference in the goodness of the match could influence the results and therefore would be useful to report. An additional criteria based on the goodness of the match could also help to exclude participants where this procedure is not working reliably.

We applied the RT-matching procedure successfully to the pilot data. Differences in RTs between correct and erroneous trials were equivalent ($M = 1$ ms, $SD = 1$ ms), i.e., they were larger than -2 ms, $t(43) = 11.16$, $p < .001$, $d_z = 1.68$, and smaller than 2 ms, $t(43) = -6.60$, $p < .001$, $d_z = -0.99$. Crucially, we only needed to match RTs once, without using an iterative trimming procedure as proposed as a fallback plan in the *Data analysis* section. RDs were still substantially shorter for erroneous than correct responses, $t(43) = 9.02$, $p < .001$, $d_z = 1.36$. As in the validation study, matching of RTs did not reduce the effect of accuracy on RDs, the effect was descriptively even larger. Considering that the validation study produced similar results and for practical reasons, we would rather refrain from including these analyses in the manuscript. We only explain the matching procedure after the pilot study in the sections *Data treatment* and *Data analyses*. Including this analysis beforehand might render the method section less accessible.

It is a good idea to evaluate the goodness of match within a participant because we believe that this might reduce data exclusion to a minimum if we have to resort to iterative matching. We would therefore propose to use such an evaluation if the initial matching procedure is not successful, i.e., if at least one of the two t-tests for a dependent variable is not significant. In that case, we could assess the individual ΔRT of each participant and compare it to both test values of the equivalence testing procedure. In the pilot data with its big sample, none of the 44 participants had a ΔRT equal or lower than -2 ms, however, for three participants erroneous responses were still over 2 ms slower than correct responses (2.1 ms, 3.6 ms and 8.9 ms). We originally proposed that if at least one of the equivalence tests of a dependent variable is not significant, we would trim the error data by removing

the bottom 5% for *each* participant. Instead, we could apply this trimming procedure to participants with a ΔRT that matches or exceeds the test values of the equivalence tests. We would do that in an iterative manner, restricting the analysis to participants with excessive values and only switch to the whole sample when the whole sample lies in between test values. Thank you for stimulating this approach that we included in the description of our analyses.

4. Whether the clarity and degree of methodological detail would be sufficient to replicate the proposed experimental procedures and analysis pipeline.

Some information is missing or need to be clarified:

- Please define RT and RD in the methods of the pilot data.
- Force measurement system: What type of force sensor is used? What is the range and the linearity of the loading cell? Are the force sensors calibrated (e.g. with pre-defined weights and without weight)? The sampling rate (250 Hz) is rather low. Is this the limit of the system?
- Please add the object dimensions of the custom-built keys.
- How do the arbitrary units relate to force (N)?
- Are statistical results corrected for multiple comparisons?

We included definitions of RT and RD in the pilot study.

The system uses three metal plates separated electrically and physically by rubber point rests. The outer plates are grounded. On the inner plate, there is high frequency tension that diverts to the outer plates if the distance to the outer plates decreases. The system can capture up to about 2 kg weight. The relationship of tension to pressure is only almost linear. The system calibrates itself one minute after switching it on or when no pressure is applied for this duration at any time. We do not employ any other calibration. The sampling rate of the system is at least 500 Hz but we noticed that communication with the intermediary RedLab is only reliable with 250 Hz. The threshold of 0.25 a.u. is about 250 g or 2.5 Newton. We discussed about converting these arbitrary units to other measures such as Newton or gram. However, we finally decided against it because a translation might not be perfectly accurate and add (minimal) noise to the data. As we are not interested in the absolute amount of force, we think that working with this arbitrary units is feasible. We provide the object dimensions in *Stimuli and apparatus*.

We do not control for multiple comparisons. One could consider controlling for multiple comparisons when testing the same hypothesis in the same dependent variable for unmatched and both matched datasets. However, we are already conservative in these tests because we only assume error cancellation if the analyses of all three datasets reveal significant effects of accuracy, whereas corrections for multiple comparisons would apply to situations in which a researcher would take any (unspecified) significant test from a range of tests to reject the null hypothesis. If the results of the analyses are ambiguous, we will make sure that there is strong evidence for the alternative or null hypothesis via Bayesian analyses.

5. Whether the authors provide a sufficiently clear and detailed description of the methods to prevent undisclosed flexibility in the experimental procedures or analysis pipeline.

This is the case. I have no further comments that could improve this point.

6. Whether the authors have considered sufficient outcome-neutral conditions (e.g. absence of floor or ceiling effects; positive controls; other quality checks) for ensuring that the results obtained are able to test the stated hypotheses.

This is the case. I don't see any further control condition that need to be added. Overall, this is a scientifically strong proposal that would make a valuable contribution to the literature.

Thank you for this positive assessment!

Reviewer 3

The study seems well designed, interesting and the paper is well written. I do have one suggestion regarding the embedding in the literature (and the clarification of writing regarding the extension the present study provides, of course fully acknowledging the value of replications). I know this might not be the main focus of the review process in this journal, nevertheless I think with a few additional clarifications the introduction of the experiment, the paper could clearly benefit. Many previous studies have used response force keys to investigate inhibiting and correcting error responses (some few examples at the end of the paper, various of these are already cited within the paper already). I think the current study could highlight some of the previous literature more, and state more explicitly what the contribution the current study aims to make. From my initial reading the contribution is the matching of RTs and PF across correct and error trials, with the addition of the distributional analyses, which I regard as very interesting.

Roger, C., Núñez Castellar, E., Pourtois, G., & Fias, W. (2014). Changing your mind before it is too late: the electrophysiological correlates of online error correction during response selection. *Psychophysiology*, 51(8), 746-760.

Gehring, W. J., & Fencsik, D. (1999, April). Slamming on the brakes: An electrophysiological study of error response inhibition. In annual meeting of the Cognitive Neuroscience Society, Washington, DC.

Hochman, E. Y., Milman, V., & Tal, L. (2017). Evidence for aversive withdrawal response to own errors. *Acta psychologica*, 180, 147-154.

Thank you for this positive feedback and for pointing us to these highly relevant papers. It is nice to see that other authors thought of error cancellation (or withdrawal) as a central part of error processing, too – and we have to admit that we did not have these papers and posters on the radar. We have incorporated these studies into the introduction and emphasize more strongly that our novel contribution to investigating error cancellation is the approach to RDs while matching for RTs and PFs.

We also elaborate more thoroughly on the necessity of these matching procedures as requested by Reviewer 1 (see their first point). In a nutshell, the present matching procedure allows for comparing active error cancellation to differences in how error responses are generated more convincingly than studies that only address the unmatched force profile or duration of the response. We therefore hope that the present study will be informative for theorizing on human error processing though we fully agree that this strategy needed to be spelled out more clearly, as we have done in the revision.

I could not find some of the Figures referred to within the manuscript on the linked-to OSF upload.

We did not upload figures to OSF and we double-checked the manuscript to ensure that we did not refer to figures on OSF in the text. What we have uploaded is the R syntax used for all analyses. This does not include any plots, however, because we created the final versions of these plots in Excel rather than R. All figures referred to within the manuscript can be found in the main manuscript or in the supplementary material to the manuscript.

I think the justification of sample size seems valid. I have only minor comments on the participant exclusion criterion and data selection procedures. The 60% correct criterion for inclusion seems lenient. It is a relatively simple task so performing almost 40% errors is almost guessing. I guess this is related to the strict 600 ms deadline which seems to add some speed-accuracy tradeoff. The requirement of at least 10 trials per cell seems valid for the main analyses. However, I wonder if this is a problem for the distributional analyses. For example, does this mean that some participants could have as few as 2-3 trials per bin in the distributional analysis here? If that is the case this seems somewhat problematic, how will the authors deal with this potential issue?

Yes, we intended to make the experiment difficult and provoke commission errors by pressuring participants to respond within a short time-window. We agree that the inclusion criterion based on accuracy is lenient, but it seems appropriate given our intention to provoke many errors. Similar studies from our lab replicated typical effects of error processing such as post-error slowing or, as presented here, fast erroneous compared to correct responses. We also found typical effects of stimulus-response binding in such designs. These observations make us confident that despite of a liberal inclusion criterion, included participants respond in a systematic manner.

In the pilot study, we excluded participants if they delivered less than 10 observations in any of the cell of the distributional analyses from all statistical analyses exactly because of the concern that you also raised in your comment. We have now included this information among others on data treatment and analyses to make the pilot study more accessible.

I have a couple of minor questions regarding the force keys. I see in the method section that the force profiles are baseline corrected using the first 10 samples. Is this from the onset of fixation? I wonder if response activation builds up in both response hands during the fixed 500ms fixation interval, and whether a pre-stimulus baseline would be more appropriate?

Yes, we use the first ten measurements after onset of fixation as a baseline. We provided participants of the validation study with error-feedback if they exceeded the response-threshold during fixation and we will employ this feedback in the proposed experiment. This should prevent participants to some degree from pre-activation. In Figure S3A in the supplementary material, you can see that until about 300 ms after target onset, force values appear to be 0 or close to 0. If response activation builds up from fixation, we should already see positive force values from target onset. As such, we would argue that response activation during fixation should only play a minor role.

„After the response has been registered, the screen will be black for an inter-trial interval of 1000 ms while the force on both keys will still be measured. The offset of the response is registered when the force matches or exceeds the threshold for the last time.“ Does the blank screen begin when the participant first exceeds the criterion, or only when they reduce the force again?

It begins when the force first exceeds the criterion. We clarified this in the text: “After the response onset has been registered...”.

„At the end of each block, participants will receive a summary on their performance with the mean RT of correct responses and the number of commission and omission errors. They will also be urged to respond as quickly as possible while trying to avoid high numbers of errors.“ Is the between block feedback performance dependent? For example, if a VP makes too few errors/too many errors, are they asked to respond faster/slower, respectively?

The mean RT, and the number of commission error reflect participants' actual performance. However, we did not include an evaluation of this performance relating to the level of errors and response time. We rather expect participants to be motivated to improve or at least maintain their performance across the experiment, and experience with (many) previous studies suggests that this is indeed the case. We therefore believe it is most promising to urge participants after each block to respond as quickly as possible while trying to avoid high numbers of errors, without explicitly judging their behavior as good or bad.

„We will only select trials for further analyses with a correct response in the preceding trial to control for potential effects of post-error processing.“ I think the selection criterion of appropriate error sequences should be made more explicit. From the plots, it seems that the sequence CCEC is required, but from sentence above it seems other sequences would be included, such as ECEC and ECEE, which of course would be influenced by differences in pre-error speeding (Dudschig, C., & Jentzsch, I. (2009). Speeding before and slowing after errors: Is it all just strategy?. *Brain research*, 1296, 56-62).

In the pilot analyses, we indeed plotted CCECC sequences. These plots suggest that there is pre-error speeding (see Figure 1B). We now emphasize in the section of the pilot study that we only selected these sequences for visualization purposes.

In the analyses of the proposed experiment, we will only look at correct vs. erroneous responses that were preceded by a correct response. Following your comment, we also briefly discuss pre-error speeding in the revised *Introduction*.

I wonder how the distribution of error responses across the experiment looks? For example, I would predict that more errors are committed at the beginning of the experiment as participants learn to S-R mapping (Of course, it is also conceivable that more errors are made towards the end of the experiment as tiredness/boredom sets in, and that these practice/tiredness effects could be different across participants). Related to this, how does a participant's force profile change across the duration of the experiment. From personal experience, I have often observed that the force profiles are extremely large to begin with and with experience/practice, participants become more efficient using only just enough force to trigger the response. The authors might want to take this into consideration.

It is true that such effects are theoretically possible, and the literature on error processing contains a few speculations on how such differences in error likelihood across time might bias experimental results. To our knowledge, however, there is relatively little danger from such effects, and we actually have just completed a methods study on this point (Pfister & Foerster, in revision).

Whether or not there is a systematic imbalance of errors across the experiment still needs to be checked for every single dataset of course. We have therefore re-analyzed the percentage of commission errors as well as PFs for the unmatched data of the validation study as suggested. In all following plots, you see the experimental blocks on the x-axis. The mean of all participants within a block is depicted in orange for errors and black for correct responses. Individual values of each participant are depicted in transparent colors. Dashed lines indicate means across participants and blocks. In the first plot, you can see that commission error rates fluctuated but showed no consistent trend across the experiment.

The second plot depicts PFs of erroneous responses and the third plot depicts PFs of correct responses. There also does not seem to be a consistent trend, except maybe that participants apply higher forces in the very first block (although this seems mostly be driven by one participant). Considering the absence of any trend, we would opt against including temporal analyses of this kind into our analyses pipeline.

The authors talk about the “successful cancelation of erroneous responses”. I wonder if there is a distinction between such successful cancelation and error correction (i.e., the subsequent execution of the correct response). I think this issue is touched upon within the proposed analyses, but I think it could be elaborated/discussed further. For example, is such error cancelation only observed on a subset of trials where the initiation of the correct response is activated? Or is this cancellation process regarded as independent of the correction process? How could this be looked into? I think this would also be a really nice aspect to analyze / present more clearly to the reader.

We agree that this is an interesting aspect to explore. Different scenarios seem valid. Agents could commit an error but detect it only after responding so that they would not be able to cancel the error but they would still be able to correct it. Agents could also detect an error early, cancel it but do not correct it because they are not able to or they do not see an advantage in correction. Agents detect an error early, cancel and correct it efficiently. With this in mind, we would argue that cancellation and correction should at least be partly independent. Both processes might still interact, of course, and this is an intriguing possibility. Hypothesis 6 approaches this relation by correlating Δ AUCs of correct and erroneous responses with erroneous subthreshold responses in correct trials. A positive correlation would indicate that a stronger attenuation of force in erroneous trials relates to the successful abortion and correction of erroneous subthreshold responses. We emphasized the correction aspect more explicitly in the revision. A convincing test of this relation, however, calls for a study that urges participants to correct their errors. We will include this consideration in the discussion of the article because it goes beyond the current aim of the study

Please comment explicitly on each of the following points in your comments to the authors:

The scientific validity of the research question(s): valid

The logic, rationale, and plausibility of the proposed hypotheses: good

The soundness and feasibility of the methodology and analysis pipeline (including statistical power analysis where applicable): good

Whether the clarity and degree of methodological detail would be sufficient to replicate exactly the proposed experimental procedures and analysis pipeline: in general yes, see minor details above

Whether the authors provide a sufficiently clear and detailed description of the methods to prevent undisclosed flexibility in the experimental procedures or analysis pipeline: yes, see minor details above

Whether the authors have considered sufficient outcome-neutral conditions (e.g. positive controls) for ensuring that the results obtained are able to test the stated hypotheses: yes, taken care of

Thank you for your kind and thoughtful review!

Appendix C

Error cancellation: Cover letter

Dr. Anna Foerster

Department for Cognitive Psychology, University of Würzburg

Röntgenring 11, 97070 Würzburg, Germany

Tel. +49 931 318901, Email: anna.foerster@uni-wuerzburg.de

Full submission of RSOS-210397 – *Error cancellation*

Dear Dr. Chambers,

Thank you very much for approving of our Stage 1 Registered Report. We have successfully completed our study in the meantime and updated our manuscript to include the results and their discussion. You will see that the findings are straightforward – error cancellation emerges already early during the commission of an error as we hypothesized.

We provide the URL of the publicly accessible online repository (osf.io/5v9es) that stores the raw data and analysis code for the pilot analyses, the validation study and the preregistered main study as well as the Stage 1 protocol (this link appears on page six in the manuscript). We did not collect data for the pre-registered study other than the pilot and validation data already included at Stage 1 prior to the date of IPA.

We have executed and analysed the experiment as originally approved, with one exception that we have discussed with Senior Publishing Editor Andrew Dunn before data collection started. We wanted to have the same arrangement of the distractors relative to the targets on screen in our preregistered main study as in the pilot and validation study but did not describe this aspect correctly in the method section. We used the intended arrangement and therefore changed the method section in this regard from Stage 1 to Stage 2. We made this change explicit for the reviewers and readers in the manuscript. Further, we highlighted all changes and additions in the manuscript and the supplement in the submitted files. We also provide versions of these documents without tracked changes to facilitate readability.

We are looking forward to your assessment of the revised version of our manuscript.

With best regards,

Anna Foerster

On behalf of the authors

Appendix D

Error cancellation: Cover letter and response to reviewers

Dr. Anna Foerster

Department for Cognitive Psychology, University of Würzburg

Röntgenring 11, 97070 Würzburg, Germany

Tel. +49 931 318901, Email: anna.foerster@uni-wuerzburg.de

Revision of RSOS-210397.R2 – *Error cancellation*

Dear Dr. Chambers,

Thank you very much for inviting a minor revision of our Stage 2 Registered Report. We prepared a point-by-point response to each suggestion of the two reviewers below. Again, we want to thank everyone involved in the review process for the careful assessment of our manuscript and the constructive comments that truly elevated our research approach and its communication.

With best regards,

Anna Foerster

On behalf of the authors

Reviewer 1

This is a Stage 2 report of a previously approved study protocol. The authors found an early error cancellation process, as they initially predicted. The introduction, rationale and hypotheses are the same as the Stage 1 report, except for some minor textual changes (e.g., changes in the tenses). The authors also closely followed the registered procedure, with one change transparently reported. The matching procedures were successful, thus the data are able to address the stated hypotheses. The conclusions are also justified given the data. Overall, I find this Stage 2 report a solid, rigorous and interesting piece of work. I have only a few minor comments, and I hope they will be helpful in preparing the paper for publication.

Thank you for this positive assessment!

The raw data files are shared publicly on OSF. However, it would be helpful to future researchers if the authors could provide a codebook, explaining the meaning of the variables in the data files.

Thank you for spotting this. We added codebooks for the pilot data, the validation data, and the main analyses on OSF.

Abstract: Now that the authors have conducted the planned experiment, it would be good to also include the main findings in the abstract, rather than only the hypothesis.

We agree and have revised the abstract accordingly.

Figure 2 shows the results based on the unmatched data and the PF-matched data. Since the authors also conducted analyses on the RT-matched data, I think it would be informative to show those results in the figure as well.

We have added the RT-matched trials as panels E and F in Figure 2 of the main manuscript and in Figure S3 the Supplementary Material.

Page 16, line 53: The text says AUCs but the subscript is PF.

Fixed.

Figure 3: I find Figure 3 a bit difficult to understand, maybe because it shows the difference between two change scores over time. More importantly, it's unclear to me what information Figure 3 conveys. The authors mentioned that the gradients for the force profile were plotted, but this figure was never discussed in the text.

In Figure 3 and Figure S4, we intended to provide a different visual perspective on the force profile for the reader. Considering your feedback about their complexity, they might not serve that purpose and we decided to delete them from the main manuscript and the Supplementary Material.

Reviewer 2

In this study, Foerster and colleagues showed that active error cancellation occurs much earlier than previous studies suggested. They followed a new approach by examining force profiles of button press responses and took great care in matching the correct and erroneous responses by response time and peak forces to provide a valid comparison of the two response types. The results of the main experiment match the ones of the validation study and confirm the pre-defined hypotheses. One of the main results is that erroneous responses led to shorter response durations within the first 100ms after movement onset. Based on the clear results, the discussion is straight-forward and concise and conveys a clear message.

Overall, this is a very convincing study. I only have some small recommendations to improve the result section and the figures.

Thank you for this kind evaluation!

Results: The authors described the hypotheses in very great detail at the of the introduction. In total, they hypothesized eight different outcomes. As I've read the result section, I missed a link between the hypotheses and the single results reported there. I suggest to add the hypotheses to the results to provide the reader a better guidance.

We now highlight how the reported statistics relate to the hypotheses in the results section.

Figure 2B, 2D: The values on the y-axis are floating in the air. I would add ticks to indicate where exactly the values are located along the axis (same for Figure S3). In addition, the error bars are hard to see. I suggest to use full instead of half error bars (also to make it consistent across the bars).

Figure S3B, S3D: Please add error bars.

We added ticks on the y-axis of the RD plots, and we changed the colors of the error bars, using only black on white or vice versa to increase visibility in Figure 2 and Figure S3. We kept half error bars in both figures to be consistent with the employed one-tailed paired-samples *t*-tests and directional confidence intervals (which go from $-\infty$ to the upper end of the CI if the upper end is a positive number, and from $+\infty$ if it is a negative number).

Page 5, typo: RDs spreads -> RD spread

Fixed.